# Active Learning of Classifiers
# with Label and Seed Queries

**Marco Bressan**
Dept. of CS, Univ. of Milan, Italy
marco.bressan@unimi.it

**Nicolò Cesa-Bianchi**
DSRC & Dept. of CS, Univ. of Milan, Italy
nicolo.cesa-bianchi@unimi.it

**Silvio Lattanzi**
Google
silviol@google.com

**Andrea Paudice**
Dept. of CS, Univ. of Milan, Italy &
Istituto Italiano di Tecnologia, Italy
andrea.paudice@unimi.it

**Maximilian Thiessen**
Research Unit ML, TU Wien, Austria
maximilian.thiessen@tuwien.ac.at

## Abstract

We study exact active learning of binary and multiclass classifiers with margin. Given an $n$-point set $X \subset \mathbb{R}^m$, we want to learn an unknown classifier on $X$ whose classes have finite *strong convex hull margin*, a new notion extending the SVM margin. In the standard active learning setting, where only *label* queries are allowed, learning a classifier with strong convex hull margin $\gamma$ requires in the worst case $\Omega\left(1 + \frac{1}{\gamma}\right)^{\frac{m-1}{2}}$ queries. On the other hand, using the more powerful *seed* queries (a variant of equivalence queries), the target classifier could be learned in $\mathcal{O}(m \log n)$ queries via Littlestone's Halving algorithm; however, Halving is computationally inefficient. In this work we show that, by carefully combining the two types of queries, a binary classifier can be learned in time $\text{poly}(n + m)$ using only $\mathcal{O}(m^2 \log n)$ label queries and $\mathcal{O}\left(m \log \frac{m}{\gamma}\right)$ seed queries; the result extends to $k$-class classifiers at the price of a $k!k^2$ multiplicative overhead. Similar results hold when the input points have bounded bit complexity, or when only one class has strong convex hull margin against the rest. We complement the upper bounds by showing that in the worst case any algorithm needs $\Omega\left(km \log \frac{1}{\gamma}\right)$ seed and label queries to learn a $k$-class classifier with strong convex hull margin $\gamma$.

## 1   Introduction

This work investigates efficient algorithms for exact active learning of binary and multiclass classifiers in the transductive setting. Given a set $X$ of $n$ points in $\mathbb{R}^m$, our goal is to learn a function $h : X \to [k]$ belonging to some class $\mathcal{H}$. In the classic active learning framework, $h$ identifies a subset of $X$, and the algorithm learns $h$ via queries LABEL$(x)$ that return $h(x)$ for any given $x \in X$. In that case, it is well-known that $h$ can be learned with $\mathcal{O}(\log n)$ LABEL queries if the *star number* of $\mathcal{H}$ is finite [Hanneke and Yang, 2015]. Unfortunately, even simple families such as linear classifiers have unbounded star number, in which case $\Omega(n)$ LABEL queries are needed in the worst case. To bypass this lower bound, it has become increasingly common to introduce *enriched queries*, that reveal additional information on $h$ and are plausible in practice. One notable example is that of *comparison* queries for linear separators in $\mathbb{R}^m$ which, given any pair of points $x, y \in X$, reveal which one is

closer to the decision boundary. As proven by Kane et al. [2017], under some margin assumptions the combination of LABEL and comparisons yields exponential savings, allowing one to learn linear separators with only $\mathcal{O}(\log n)$ queries.

In this work we combine LABEL queries with *seed queries*. For any $U \subset X$ and any $i \in [k]$, a query SEED$(U, i)$ returns an abitrary point $x$ in $U \cap C_i$, where $C_i = h^{-1}(i)$, or NIL if no such $x$ exists. SEED queries are natural in certain settings like crowdsourcing—e.g., finding the image of a car, see also Beygelzimer et al. [2016]—and have been used implicitly or explicitly in several works [Hanneke, 2009, Balcan and Hanneke, 2012, Attenberg and Provost, 2010, Tong and Chang, 2001, Doyle et al., 2011, Bressan et al., 2021b]. It is not hard to see that, using SEED alone, one can implement Littlestone's Halving algorithm and learn any $h \in \mathcal{H}$ with $\mathcal{O}(\log |\mathcal{H}|)$ queries[1]. For instance, linear separators in $\mathbb{R}^m$ can be learned with $\mathcal{O}(m \log n)$ SEED queries. The catch is that, save for special cases, it is not known how to run the Halving algorithm in polynomial time. Therefore, using SEED to obtain a computationally efficient active learning algorithm is less trivial than it seems at first glance.

The goal of this work is understanding whether one can actively learn binary and multiclass classifiers efficiently by using LABEL and SEED queries together. In line with Kane et al. [2017] and other previous works, we make assumptions on $\mathcal{H}$. Our main assumption is that every class $C_i$ has *strong convex hull margin* $\gamma > 0$. This means that, for any $j \neq i$, $C_i$ and $C_j$ are linearly separable with a margin that is at least $\frac{\gamma}{2}$ times the diameter of $C_i$. Moreover, it is sufficient that this hold under some pseudometric $d_i$, unknown to the learner, that is homogeneous and invariant under translation (i.e., induced by a seminorm). This gives to every class its own personalized notion of distance that can be sensitive to the "scale" of the class. This assumption strictly generalizes the classical SVM margin; and, when suitably generalized, it captures stability properties of center-based clusterings Awasthi et al. [2012], Bilu and Linial [2012].

Using LABEL alone, Bressan et al. [2021a] showed that learning a multiclass classifier with (strong) convex hull margin $\gamma > 0$ requires between $\Omega\left(1 + \frac{1}{\gamma}\right)^{(m-1)/2}$ and $\tilde{\mathcal{O}}\left(k^3 m^5 \left(1 + \frac{1}{\gamma}\right)^m \log n\right)$ queries. This exponential dependence on $m$ implies that, unless $m \ll \log n / \log \frac{1}{\gamma}$, one needs $\Theta(n)$ LABEL queries in the worst case. On the other hand our margin implies linear separability and thus, as noted above, a $\mathcal{O}(m \log n)$ SEED query bound for the binary case, but with a running time that can be superpolynomial. This leaves open the following problem, which is the subject of this work:

> Can one learn a multiclass classifier $h$ with strong convex hull margin $\gamma > 0$ on $X \subset \mathbb{R}^m$ in time $\mathrm{poly}(n+m)$ using a number of queries that grows *polynomially* with $m$?

We solve the above question in the affirmative by proving that, with a careful combination of LABEL and SEED queries, one can do much better than using either query in isolation. For binary classification ($k = 2$), we show:

**Theorem 1.** *Any binary classifier $h$ with strong convex hull margin $\gamma > 0$ over $X \subset \mathbb{R}^m$ can be learned in time $\mathrm{poly}(n+m)$ using in expectation $\mathcal{O}(m^2 \log n)$ LABEL queries and $\mathcal{O}\left(m \log \frac{m}{\gamma}\right)$ SEED queries.*[2]

Note that, unless $\gamma$ is exceedingly small, Theorem 1 uses far fewer SEED than LABEL queries, which is a strength since SEED is arguably more expensive to implement. For instance, if $\gamma = \Omega(1/\mathrm{poly}(m))$ then we use $\mathcal{O}(m^2 \log n)$ LABEL queries but only $\mathcal{O}(m \log m)$ SEED queries. To prove Theorem 1 we design a novel algorithm that works in two phases. The first phase learns what we call an $\alpha$-*rounding* of $X$ w.r.t. $h$. Loosely speaking, this is a partition $(X_1, X_2)$ of $X$ such that each $X_i$ lies inside $\alpha \, \mathrm{conv}(C_i)$ where $\mathrm{conv}(C_i)$ is the convex hull of $C_i$ (see below for the formal definition). We show that, in polynomial time and using $\mathcal{O}(m^2 \log n)$ LABEL queries, one can compute an $\alpha$-rounding of $X_i$ for $\alpha = \mathcal{O}(m^3)$. This allows us to put $X_i$ in near-isotropic position so that $X_i$ has radius 1 and to separate $C_1 \cap X_i$ from $C_2 \cap X_i$ with margin $\eta = \Omega(\gamma/m^3)$. In the second phase, the algorithm uses SEED to implement a cutting plane algorithm that learns $C_1 \cap X_i$ and $C_2 \cap X_i$ using $\mathcal{O}\left(m \log \frac{1}{\eta}\right) = \mathcal{O}\left(m \log \frac{m}{\gamma}\right)$ queries in time $\mathrm{poly}(n + m)$.

---

[1]Halving uses equivalence queries (testing if a given subset of $X$ coincides with the target concept) each of which can be simulated using two SEED queries.

[2]This running time as well as those of Theorem 2 and 3 are actually in high probability as implied by Theorem 10; we have omitted this fact to keep the statements light.

Using a recursive approach, Theorem 1 can be extended to $k > 2$ at the price of a $k!k^2$ multiplicative overhead:

**Theorem 2.** *Any $k$-class classifier $h$ with strong convex hull margin $\gamma > 0$ over $X \subset \mathbb{R}^m$ can be learned in time $\mathrm{poly}(n + m)$ using in expectation $\mathcal{O}(k!\,k^2\,m^2 \log n)$ LABEL queries and $\mathcal{O}(k!\,k^2\,m \log \frac{m}{\gamma})$ SEED queries.*

We also consider the case where only one class has strong convex hull margin against the rest of the points w.r.t. a metric $d$ induced by a norm $\|\cdot\|_d$. In this case we obtain a bound parameterized by the distortion $\kappa_d$ of $d$ (see Section 1.1):

**Theorem 3.** *Suppose $C \subset X$ has strong convex hull margin $\gamma \in (0, 1]$ w.r.t. a metric $d$ with distortion $\kappa_d < \infty$. Given only $X$, one can learn $C$ in time $\mathrm{poly}(n + m)$ using $\mathcal{O}(\log n)$ LABEL queries and $\mathcal{O}(m \log \frac{\kappa_d}{\gamma})$ SEED queries in expectation.*

As an application of our cutting-plane algorithm we also show that one can learn a $k$-class classifier whose classes are pairwise linearly separable in time $\mathrm{poly}(n + m)$ using, in expectation, $\mathcal{O}(k^2 m^3 B)$ SEED queries if every $x \in X$ has rational coordinates that can be encoded in $B$ bits, and $\mathcal{O}(k^2 m(B + m \log m))$ SEED queries if every $x \in X$ lies on the grid over $[-1, 1]^m$ with stepsize $2^{-B/m}$. It should be noted that, unlike most previous algorithms, all our algorithms do not need knowledge of $\gamma$. Moreover, all the bounds above can be turned from expectation to high probability.[3]

Finally, we show that the algorithms of Theorem 1 and 2 are nearly optimal:

**Theorem 4.** *For all $m \geq 2$, all $k \geq 2$, and all $\gamma \leq m^{-3/2}/16$ there exists a distribution of instances with $k$ classes in $\mathbb{R}^m$ with strong convex hull margin $\gamma$ where any randomized algorithm using SEED and LABEL queries that returns $\mathcal{C}$ with probability at least $\frac{1}{2}$ makes at least $\lfloor \frac{k}{2} \rfloor \frac{m}{24} \log \frac{1}{2\gamma}$ total queries in expectation.*

### 1.1 Preliminaries and notation

The input to our problem is a pair $(X, k)$, where $X \subset \mathbb{R}^m$ and $k \in \mathbb{N}$ with $2 \leq k \leq n = |X|$. The algorithm has access to oracles $O_{\text{LABEL}}$ and $O_{\text{SEED}}$ which provide respectively LABEL and SEED queries. The oracles $O_{\text{LABEL}}, O_{\text{SEED}}$ behave consistently with some target classifier $h : X \to [k]$. For any $x \in X$, LABEL$(x)$ returns $h(x)$. For any $U \subseteq X$ and any $i \in [k]$, SEED$(U, i)$ returns an abitrary element $x \in U \cap C_i$ if $U \cap C_i \neq \emptyset$, and NIL otherwise, where $C_i = h^{-1}(i)$. We often think of $h$ as of the partition $\mathcal{C} = (C_1, \ldots, C_k)$ and we call each $C_i$ a *class* or *cluster*.

A pseudometric is a symmetric and subadditive function $d : \mathbb{R}^m \times \mathbb{R}^m \to \mathbb{R}_{\geq 0}$ such that $d(x, x) = 0$ for all $x \in \mathbb{R}^m$; unlike a metric, $d(x, y)$ can be 0 for $x \neq y$. In this work $d$ is always induced by a seminorm and thus homogeneous and invariant under translation: $d(u + ax, u + ay) = |a|\, d(x, y)$ for all $x, y, u \in \mathbb{R}^m$ and all $a \in \mathbb{R}$. For a pseudometric $d$ and a set $A \subset \mathbb{R}^m$, we let $\phi_d(A) = \sup\{d(x, y) : x, y \in A\}$ denote the diameter of $A$ under $d$. For $x \in \mathbb{R}^m$ and $r \geq 0$ we denote by $B_d^m(x, r)$ and $S_d^{m-1}(x, r)$ respectively the closed ball and the hypersphere with center $x$ and radius $r$ in $\mathbb{R}^m$ under $d$. When $d$ is omitted we assume $d = d_{\text{euc}}$ where $d_{\text{euc}}$ is the Euclidean metric. We may also omit the superscript if clear from the context. The distortion of a (pseudometric) $d$ is $\kappa_d = \sup_{u,v \in S^{m-1}(0,1)} \|u\|_d / \|v\|_d$.

For any set $A \subset \mathbb{R}^m$, any $\mu \in \mathbb{R}^m$, and any $\lambda > 0$, let $\sigma(A, \mu, \lambda) = \mu + \lambda(A - \mu)$ be the scaling of $A$ about $\mu$ by a factor of $\lambda$. For two sets $A, B \subset \mathbb{R}^m$, we write $A \leq \lambda B$ if $A \subseteq \sigma(B, z, \lambda)$ for some $z \in \mathbb{R}^m$. We may use $x$ in place of $A$ if $A = \{x\}$. If $A$ is bounded, then $\text{MVE}(A)$ denotes the minimum-volume enclosing ellipsoid (MVEE, or Löwner-John ellipsoid) of $A$. Our proofs repeatedly use John's theorem; that is, $\sigma(E, \mu, 1/m) \subseteq \text{conv}(A)$ where $\mu$ is the center of $E = \text{MVE}(A)$ and $\text{conv}(A)$ is the convex hull of $A$. Given $A, B \subseteq \mathbb{R}^m$, we say that $A$ and $B$ are linearly separable with margin $r$ if there exist $u \in S^{m-1}(0, 1)$ and $b \in \mathbb{R}$ such that $\langle u, x \rangle + b \leq -r$ for all $x \in A$ and $\langle u, x \rangle + b \geq r$ for all $x \in B$.

We consider classifiers satisfying the following property:[4]

---

[3]Formally, for some universal constant $a > 0$, each one of our bounds in the form $\mathbb{E}[Q] \leq q$, where $Q$ is the number of queries, implies $\Pr(Q \geq q + \epsilon q) \leq \exp(-a\epsilon q)$ for all $\epsilon \geq 0$.

[4]Actually, all our upper bounds hold under a weaker condition: that for every $i$ and every $j \in [k] \setminus \{i\}$ there is a $d_{ij}$ giving the margin.

**Definition 5.** *A class $C_i$ has strong convex hull margin $\gamma > 0$ if there exists a pseudometric $d_i$ induced by a seminorm over $\mathbb{R}^m$ such that $d_i(\mathrm{conv}(C_j), \mathrm{conv}(C_i)) > \gamma \, \phi_{d_i}(C_i)$ for all $j \in [k] \setminus \{i\}$. If this holds for all $i \in [k]$ then we say $\mathcal{C}$ has strong convex hull margin $\gamma$.*

**Remarks.** The margin of Definition 5 captures natural scenarios that SVM margin does not. For instance, suppose we are clustering fruits on the basis of weight and colour. First, a fruit weighting more than, say, $1.5$ times the typical weight of a species probably does not belong to it; but the typical weight varies greatly across species. Our margin captures this scenario, as it is expressed as a fraction of the class' diameter. Second, different fruit species have different separating features; for instance, weight does not separate well oranges from bananas, but colour does. Our margin captures this aspect, too, by allowing the metric that determines the margin to be a function the class. It is also known that the SVM margin $\gamma_{\mathrm{SVM}}$ can be arbitrarily smaller than $\gamma$; for instance there are simple cases with $\gamma > 1$ but $\gamma_{\mathrm{SVM}} < e^{-n}$ (see Bressan et al. [2021a]). Hence a large $\gamma$ does not imply good bounds for standard algorithms based on SVM margin (e.g., the Perceptron).

## 2 Related work

It is well known that active learning may achieve exponential savings in label complexity. That is, there are natural concept classes that can be learned with a number of LABEL queries exponentially smaller than that of passive learning. Hanneke and Yang [2015] characterize the label complexity of concept classes in terms of their star number. However, the star number of many natural classes such as linear classifiers is unbounded, implying a strong lower bound of $\Omega(n)$ LABEL queries.

This and other negative results motivated research on enriched queries. Kane et al. [2017] prove that active learnability is characterized by the inference dimension of the concept class $\mathcal{H}$ under the set of allowed queries $\mathcal{Q}$, as long as those queries are local (i.e., are a function of a constant number of instances). This yields exponential savings when $\mathcal{H}$ is the class of linear separators and $\mathcal{Q}$ contains label queries and comparison queries (which, given two points, reveal which one is closer to the decision boundary), provided the classes have SVM margin or bounded bit complexity. Hopkins et al. [2020] give similar results under distributional assumptions. Unfortunately, bounded inference dimension does not automatically yield efficient algorithms, although it implies active learning algorithms with bounded memory [Hopkins et al., 2021].

SEED and their variants are motivated and used by Hanneke [2009] as *positive example queries*, by Balcan and Hanneke [2012] as *conditional class queries*, and by Beygelzimer et al. [2016], Attenberg and Provost [2010] as *search queries*. They are also used implicitly by Tong and Chang [2001], Doyle et al. [2011], and Vikram and Dasgupta [2016]. SEED queries have been used in cluster recovery [Bressan et al., 2021b] and yield exponential savings in non-realizable learning settings [Balcan and Hanneke, 2012]. It also easy to see that SEED queries are equivalent to *partial equivalence* queries of Maass and Turán [1992] and to *subset* plus *superset* queries of Angluin [1988]. To the best of our knowledge, no work combines LABEL and SEED as we do here.

Little is known about the SEED complexity of learning a concept class $\mathcal{H}$ actively in polynomial time. On the one hand, the inference dimension lower bounds of Kane et al. [2017] are inapplicable, as SEED queries are not local. On the other hand the Littlestone dimension of $\mathcal{H}$ yields an upper bound, but not necessarily an efficient algorithm; in fact, it is well known that (some sub-problem solved by) Halving is hard in general, see Gonen et al. [2013]. For $k = 2$, we can use SEED to emulate *equivalence* queries, for which polynomial-time algorithms are known in some special cases. In particular, the algorithm of Maass and Turán [1994] could replace our cutting-planes subroutine under an implicit discretization of the space through a grid with step-size $\mathcal{O}(\gamma/m^4)$. However, this gives a polynomial-time algorithm that uses $\mathcal{O}(m^2 \log{^m/_\gamma})$ SEED queries, which is $\mathcal{O}(m)$ times our bound. Moreover, Maass and Turán [1994] use *proper* equivalence queries (i.e., the queried concept must be in the class), for which they show a lower bound of $\Omega(m^2 \log{^m/_\gamma})$. Finally, these techniques do not seem to extend to the case $k > 2$.

Our notion of margin strengthens the convex hull margin of Bressan et al. [2021a] by requiring $d(\mathrm{conv}(C_j), \mathrm{conv}(C_i)) > \gamma\phi(C_i)$ rather than $d(C_j, C_i) > \gamma\phi(C_i)$. It is not hard to see that the convex hull margin can be arbitrarily smaller than our strong convex hull margin. Finally, the polytope margin of Gottlieb et al. [2018] assumes that each class is in the intersection of a finite number of halfspaces with margin. It is easy to see that this condition is strictly stronger than ours.

# 3  Upper Bounds

This section gives the proofs of Theorem 1 and Theorem 2. The algorithm behind both theorems has two phases which are described in the next subsections. The case $k > 2$ is essentially the same as for $k = 2$, except for an adaptation in the second phase.

## 3.1  The First Phase: Rounding the Classes

The first phase of our algorithms learns what we call an $\alpha$-rounding of $X$.

**Definition 6.** *An $\alpha$-rounding of $X$ (w.r.t. $h$) is a sequence of pairs $((X_i, E_i))_{i \in [k]}$ where $(X_i)_{i \in [k]}$ is a partition of $X$, and where $E_i$ for $i \in [k]$ is an ellipsoid such that $X_i \subseteq E_i$ and $E_i \leq \alpha \operatorname{conv}(C_i)$.*

The idea is that, if $((X_i, E_i))_{i \in [k]}$ is an $\alpha$-rounding of $X$, then $E_i$ gives an approximation of the pseudometric $d_i$ witnessing the strong convex hull margin of $C_i$. Indeed, let $p_i$ be the pseudometric induced by $E_i$, the one such that $E_i = B_{p_i}(\mu_i, 1)$ where $\mu_i$ is the center of $E_i$; we prove:

**Lemma 7.** *If $((X_i, E_i))_{i \in [k]}$ is an $\alpha$-rounding of $X$ then $p_i(\operatorname{conv}(X_i \cap C_i), \operatorname{conv}(X_i \cap C_j)) \geq \frac{\gamma}{\alpha}$ for all distinct $i, j \in [k]$.*

*Proof.* If $\mu_i$ is the center of $E_i$, then $E_i = B_{p_i}(\mu_i, 1)$. Let $d_i$ be any pseudometric witnessing that $C_i$ has strong convex hull margin $\gamma > 0$. As the margin is invariant under scaling, we can assume $\phi_{d_i}(C_i) = 1$ and $\operatorname{conv}(C_i) \subseteq B_{d_i}(z_i, 1)$ for some $z_i \in \mathbb{R}^m$. Therefore:

$$B_{p_i}(\mu_i, 1) = E_i \leq \alpha \operatorname{conv}(C_i) \subseteq \alpha B_{d_i}(z_i, 1) \tag{1}$$

As $p_i$ and $d_i$ are homogeneous and invariant under translation this implies $p_i \geq \frac{d_i}{\alpha}$ and thus $p_i(\operatorname{conv}(X_i \cap C_j), \operatorname{conv}(X_i \cap C_i)) \geq \frac{1}{\alpha} d_i(\operatorname{conv}(X_i \cap C_j), \operatorname{conv}(X_i \cap C_i))$. Moreover, by monotonicity under taking subsets and by the margin assumption $d_i(\operatorname{conv}(X_i \cap C_j), \operatorname{conv}(X_i \cap C_i)) \geq d_i(\operatorname{conv}(C_j), \operatorname{conv}(C_i)) \geq \gamma \phi_{d_i}(C_i) = \gamma$. Combining the two inequalities yields the thesis. $\square$

We will use Lemma 7 in the second phase. First, we show how to compute an $\alpha$-rounding of $X$ efficiently. We sample points independently and uniformly at random from $X$ until we find $\Theta(m^2)$ points $S_i$ with the same label $i$. As the VC dimension of ellipsoids in $\mathbb{R}^m$ is $\mathcal{O}(m^2)$, by standard generalization error bounds with constant probability the MVE of $S_i$ contains at least half of $C_i$. We then store that MVE together with the index $i$, remove $S_i$ from $X$, and repeat until $X$ becomes empty. At that point for each $i \in [k]$ we "merge" together all points in the MVEs that were computed for class $i$, and compute the MVE of this merged set. We show that this produces an $\alpha$-rounding of $X$ after $\mathcal{O}(k \log n)$ rounds in expectation.[5] The resulting algorithm $\mathrm{Round}$ is listed below; Figure 1 depicts its behaviour on a toy example.

**Lemma 8.** $\mathrm{Round}(X, k)$ *returns an $m^2(m + 1)$-rounding of $X$ in time $\operatorname{poly}(n + m)$ using $\mathcal{O}(k^2 m^2 \log n)$ LABEL queries in expectation.*

*Proof sketch.* First we show that $E_i \leq m^2(m + 1) \operatorname{conv}(C_i)$ for all $i \in [k]$. This is trivial if $E_i = \emptyset$, so let $E_i \neq \emptyset$ and let $\ell_i \geq 1$ be the value of $h_i$ at return time. For every $h = 1, \ldots, \ell_i$ let $E_i^h = \mathrm{MVE}(S_i^h)$ and let $\mu_i^h$ be the center of $E_i^h$. Using John's theorem one can show that $\sigma\left(E_i, \mu_i, \frac{1}{m}\right) \subseteq \operatorname{conv} \bigcup_{h=1}^{\ell_i} \sigma\left(\operatorname{conv}(S_i^h), \mu_i^h, m\right)$ and $\sigma\left(\operatorname{conv}(S_i^h), \mu_i^h, m\right) \subseteq \sigma(\operatorname{conv}(C_i), \mu, m(m + 1))$. By taking the union over all $h \in [\ell_i]$ we conclude that $\sigma\left(E_i, \mu_i, \frac{1}{m}\right) \subseteq \sigma(\operatorname{conv}(C_i), \mu, m(m + 1))$, that is, $E_i \leq m^2(m+1) \operatorname{conv}(C_i)$. It is also easy so see that $(X_i)_{i \in [k]}$ is a partition of $X$, hence $((X_i, E_i))_{i \in [k]}$ is an $m^2(m+1)$-rounding of $X$.

For the running time, the **for** loops perform $k \leq n$ iterations, and the **while** loop performs at most $n$ iterations as each iteration strictly decreases the size of $X$. The running time of any iteration is dominated by the computation of $\mathrm{MVE}(S_i)$ or $\mathrm{MVE}(X_i)$, which takes time $\operatorname{poly}(n + m)$, see above. Hence $\mathrm{Round}(X, k)$ runs in time $\operatorname{poly}(n + m)$. For the query bounds, the **while** loop makes $\mathcal{O}(m^2 k)$ LABEL queries per iteration. By standard generalization bounds, since the VC dimension of

---

[5]What we actually want is, given a finite set $S \subset \mathbb{R}^m$, an ellipsoid $\mathcal{E}$ such that $\frac{1}{(1+\epsilon)d} \mathcal{E} \subset \operatorname{conv}(S) \subset \mathcal{E}$. This can be computed in $\mathcal{O}(|S|^{3.5} \ln(|S|/\epsilon))$ operations in the real number model of computation, see Khachiyan [1996]. For simplicity however we just assume that we can compute $\mathcal{E} = \mathrm{MVE}(S)$ in polytime.

**Algorithm 1:** $\mathrm{Round}(X, k)$

---

**for** $i \in [k]$ **do** $h_i \leftarrow 0$
**while** $X \neq \emptyset$ **do**

    draw points independently u.a.r. from $X$ and LABEL them until for some $i \in [k]$ we draw a
    (multi)set of $cm^2$ points from $C_i$
    $h_i \leftarrow h_i + 1$
    $S_i^{h_i} \leftarrow$ the sample of $cm^2$ points from $C_i$
    $X_i^{h_i} \leftarrow X \cap \mathrm{MVE}(S_i^{h_i})$
    $X \leftarrow X \setminus X_i^{h_i}$

**for** $i \in [k]$ **do**

    $X_i \leftarrow X_i^1 \cup \ldots \cup X_i^{h_i}$ (set to $\emptyset$ if $h_i = 0$)
    $E_i \leftarrow \mathrm{MVE}(X_i)$ (set to $\emptyset$ if $X_i = \emptyset$)

**return** $((X_i, E_i))_{i \in [k]}$

---

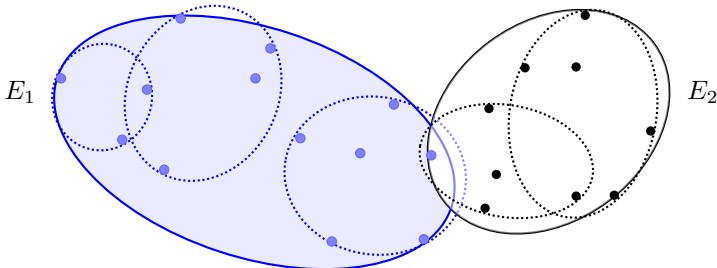

Figure 1: A toy example in $\mathbb{R}^2$ with $k = 2$; black points are in $C_1$, blue points in $C_2$. $\mathrm{Round}(X, 2)$ computes first the ellipsoids $E_2^1, E_2^2$ (dotted black, from left to right), and then the ellipsoids $E_1^1, E_1^2, E_1^3$ (dotted blue, from left to right). Finally it computes $E_1$ (solid blue) and $E_2$ (solid black). $X_1$ and $X_2$ consist of the points in the blue and white areas respectively. Note that $X_2$ contains a point of $C_1$.

ellipsoids in $\mathbb{R}^m$ is $\mathcal{O}(m^2)$, $E_i^h$ contains at least half of $X \cap C_i$ with probability at least $\frac{1}{2}$, and thus the expected number of rounds before $X$ becomes empty is in $\mathcal{O}(k \lg n)$, see Bressan et al. [2021a]. We conclude that $\mathrm{Round}(X, k)$ uses $\mathcal{O}(m^2 k^2 \lg n)$ LABEL queries in expectation. $\qquad \square$

## 3.2 The Second Phase: Finding a Separator via Cutting Planes

Let $((X_i, E_i))_{i \in [k]}$ be the output of $\mathrm{Round}(X, k)$, and fix $i \in [k]$. For each $j \in [k] \setminus \{i\}$, we want to separate $X_i \cap C_i$ from $X_i \cap C_j$. To this end, first we use $E_i$ to perform a change of coordinates; this puts $X_i$ inside the unit ball and ensures that $X_i \cap C_i$ and $X_i \cap C_j$ are linearly separated with margin $\gamma_{\mathrm{SVM}} = \Omega(\gamma m^{-3})$. Next, by calling $C_i$ the positive class $(+1)$ and $C_j$ the negative class $(-1)$, and letting $X = X_i$ for simplicity, one can reduce the task to the following problem. Consider a *partial* classifier $h : X \rightarrow \{+1, -1, *\}$. The algorithm has access to an oracle answering queries SEED$(U, y)$ where $U \subseteq X$ and $y \in \{+1, -1\}$, and its goal is to compute a separator of $X$:

**Definition 9.** *Let* $X \subset \mathbb{R}^m$ *and* $h : X \rightarrow \{+1, -1, *\}$. *A separator of* $X$ *(w.r.t.* $h$*) is a partition* $(X_+, X_-)$ *of* $X$ *such that, for every* $x \in X$*, if* $h(x) = +1$ *then* $x \in X_+$ *and if* $h(x) = -1$ *then* $x \in X_-$.

A separator of $X$ can be learned, for instance, by the Perceptron (using SEED to find counterexamples). However, this would yield a query and running time bound of $\mathcal{O}(1/\gamma_{\mathrm{SVM}}^2) = \mathcal{O}(m^6/\gamma^2)$. We provide CPLearn, a cutting-plane algorithm based on SEED that is much more query-efficient (in fact, near-optimal):

**Theorem 10.** *Let* $X \subset \mathbb{R}^m$ *and* $h : X \rightarrow \{+1, -1, *\}$, *and suppose* $h^{-1}(+1)$ *and* $h^{-1}(-1)$ *are linearly separable with margin* $r$. *Given* $X$ *and access to* SEED *for labels* $\{+1, -1\}$, CPLearn$(X)$

*computes a separator of $X$ w.r.t. $h$ using $\mathcal{O}(m \log \frac{R}{r})$ SEED queries in expectation, where $R = \max_{x \in X} \|x\|_2$, and running with high probability[6] in time $\text{poly}(m + |X|)$.*

*Proof.* (*Sketch*) First, we lift $X$ to $\mathbb{R}^{m+1}$. This reduces the problem to finding a homogeneous linear separator. To this end we let $X' = \{x' : x \in X\}$ where $x'$ is obtained by appending to $x$ an $(m+1)$-th coordinate that is equal to $R$, and we extend $h$ to $X'$ in the obvious way. It is easy to prove that $X'$ has radius at most $2R$ and that in $X'$ the two classes are linearly separable with margin $\frac{r}{2}$.

Next, we learn a separator of $X'$ w.r.t. $h$ via cutting planes—see, e.g., Mitchell [2003]. Let $V_0 = B^{m+1}(0,1)$. Every point $u \in V_0$ identifies the halfspace $H(u) = \{z \in \mathbb{R}^{m+1} : \langle u, z \rangle \geq 0\}$. For $i = 1, 2, \ldots$, $V_i$ will be our version space, and we compute $V_{i+1}$ from $V_i$ as follows. Let $\mu_i$ be the center of mass of $V_i$, and let $X'_i = X' \cap H(\mu_i)$. By issuing $\text{SEED}(X'_i, -1)$ and $\text{SEED}(X' \setminus X'_i, +1)$ we learn whether $(X'_i, X' \setminus X'_i)$ is a separator of $X'$ w.r.t. $h$, in which case we return the corresponding partition of $X$, or we obtain a point $u_i$. In the second case, we let $V_{i+1} = V_i \cap U_i$ where $U_i = \{x \in \mathbb{R}^{m+1} : h(u_i) \cdot \langle u_i, x \rangle \geq 0\}$. By [Gilad-Bachrach et al., 2004, Theorem 2] this procedure returns a separator of $X'$ w.r.t. $h$ using at most $\frac{2m}{\log \frac{e}{e-1}} \log \frac{4R}{r/2} = \mathcal{O}(m \log \frac{R}{r})$ queries.

Unfortunately, computing $\mu_i$ is hard in general [Rademacher, 2007]. We instead compute an estimate $\hat{\mu}_i$ that, used in place of $\mu_i$, ensures $\frac{\text{vol}(V_{i+1})}{\text{vol}(V_i)}$ is bounded away from 1 with high probability; the expected query bound follows by adapting the proof of [Gilad-Bachrach et al., 2004]. Assume for the moment that $V_i$ is well-rounded—that is, it contains a ball of radius $r = \text{poly}(m)$ and is contained in a ball of radius 1. To compute $\hat{\mu}_i$ we average over $\text{poly}(n + m)$ independent uniform points from $V_i$, which can be draw efficiently thanks to the rounding condition. At this point we use $\hat{\mu}_i$ in place of $\mu_i$ to invoke SEED and obtain a violated constraint $U_i$. Howewer, setting $V_{i+1} = V_i \cap U_i$ could make $V_{i+1}$ far from rounded (too "thin"), making sampling inefficient at the next round. Therefore we rotate $U_i$ so to obtain a weaker constraint $U_i^*$, one that still contains $V_i \cap U_i$ but that has $\hat{\mu}_i$ on its boundary, and let $V_{i+1} = V_i \cap U_i^*$. By the assumption on $\hat{\mu}_i$ this implies that $\text{vol}(V_{i+1}) \geq \frac{1}{3} \text{vol}(V_i)$; therefore by sampling uniform points from $V_i$ we can obtain a large sample in $V_{i+1}$, from which we can put $V_{i+1}$ in a rounding position. See the full proof for all the details. $\square$

To the best of our knowledge, CPLearn is the first efficient algorithm that achieves the query upper bound of Theorem 10, even for the special case of SVM margin.

### 3.3 Wrap-Up

We wrap up our algorithms, starting with the case $k = 2$; the case $k \geq 2$ is slightly more involved.

---

**Algorithm 2:** $\text{BinLearn}(X)$

---
$((X_1, E_1), (X_2, E_2)) \leftarrow \text{Round}(X)$
**for** $i \leftarrow 1, 2$ **do**
    change system of coordinates so that $E_i$ becomes the unit ball
    $(X_{i+}, X_{i-}) \leftarrow \text{CPLearn}(X_i)$ with $h : X_i \to \{1, 2\}$
**return** $(X_{1+} \cup X_{2-}, X_{2+} \cup X_{1-})$

---

**Theorem 11.** *Suppose $k = 2$. Then $\text{BinLearn}(X)$ returns $\mathcal{C} = (C_1, C_2)$ in time $\text{poly}(n + m)$ using in expectation $\mathcal{O}(m^2 \log n)$ LABEL queries and $\mathcal{O}(m \log \frac{m}{\gamma})$ SEED queries.*

*Proof.* By Lemma 8, $\text{Round}(X)$ runs in time $\text{poly}(n + m)$, makes $\mathcal{O}(m^2 \log n)$ LABEL queries in expectation, and returns an $\mathcal{O}(m^3)$-rounding of $X$. It is immediate to see that, after the change of coordinates, $X_i$ has radius $R \leq 1$, while $C_1 \cap X_1$ and $C_2 \cap X_1$ are separated linearly with margin $r = \Omega(\gamma m^{-3})$. By Theorem 10 then, $\text{CPLearn}(X_i)$ returns the partition of $X_i$ induced by $h$ in time $\text{poly}(|X_i| + m) = \text{poly}(n + m)$ using $\mathcal{O}(m \log \frac{R}{r}) = \mathcal{O}(m \log \frac{m}{\gamma})$ expected SEED queries. $\square$

---

[6]This means that the running time can be brought in $\text{poly}(m + |X|)$ with probability $1 - \exp(-(m + |X|))$.

For $k \geq 2$ we proceed as follows. Let $\mathbf{k} = [k]$. We take $X_i$ for each $i \in \mathbf{k}$ in turn, and for each $j \in \mathbf{k} \setminus i$, we use CPLearn to compute a separator for $i, j$ in $X_i$. By intersecting the left side of all those separators we obtain $X_i \cap C_i$. Then we recurse on $X_i \setminus C_i$, updating $\mathbf{k}$ to $\mathbf{k} \setminus i$. The resulting algorithm KClassLearn is listed below and yields:

**Theorem 12.** KClassLearn$(X, [k])$ *returns* $\mathcal{C}$ *in time* $\mathrm{poly}(n + m)$ *using in expectation* $\mathcal{O}(k!k^2\, m^2 \log n)$ LABEL *queries and* $\mathcal{O}\big(k!k^2\, m \log \frac{m}{\gamma}\big)$ SEED *queries.*

*Proof.* We adapt the proof of Theorem 11. Observe that KClassLearn$(X, [k])$ makes at most $\min(k!, n)$ recursive calls; the $n$ in the $\min$ comes from the fact that any given (recursive) call learns the label of at least one unlabeled point. Now, every (recursive) call makes one invocation to Round$(X)$, which by Lemma 8 uses time $\mathrm{poly}(n + m)$ and $\mathcal{O}(k^2 m^2 \log n)$ LABEL queries, and $\mathcal{O}(k^2)$ invocations to CPLearn$(X_i)$, each of which by Theorem 10 uses $\mathrm{poly}(n + m)$ time and $\mathcal{O}\big(m \log \frac{m}{\gamma}\big)$ SEED queries. $\qquad\square$

---

**Algorithm 3:** KClassLearn$(X, \mathbf{k})$

---

$k \leftarrow |\mathbf{k}|$
**if** $k = 1$ **then** query any point of $X$ and label all of $X$ accordingly
**else**
$\quad$ $((X_i, E_i))_{i \in [k]} \leftarrow \mathrm{Round}(X)$
$\quad$ **for** $i \in \mathbf{k}$ **do**
$\quad\quad$ change system of coordinates so that $E_i$ becomes the unit ball
$\quad\quad$ **for** $j \in \mathbf{k} \setminus i$ **do**
$\quad\quad\quad$ $(C_{ij}, \overline{C_{ij}}) \leftarrow \mathrm{CPLearn}(X_i)$ with $h : X_i \to \{i, j\}$
$\quad\quad$ $\widehat{C}_i \leftarrow \bigcap_{j \in \mathbf{k} \setminus i} C_{ij}$
$\quad\quad$ mark all of $\widehat{C}_i$ with label $i$
$\quad\quad$ **if** $X_i \setminus \widehat{C}_i \neq \emptyset$ **then** KClassLearn$(X_i \setminus \widehat{C}_i, \mathbf{k} \setminus i)$

---

## 4 Lower Bounds

This section gives a detailed sketch of the proof of Theorem 4, recalled here for convenience:

**Theorem 4.** *For all $m \geq 2$, all $k \geq 2$, and all $\gamma \leq m^{-3/2}/16$ there exists a distribution of instances with $k$ classes in $\mathbb{R}^m$ with strong convex hull margin $\gamma$ where any randomized algorithm using* SEED *and* LABEL *queries that returns* $\mathcal{C}$ *with probability at least* $\frac{1}{2}$ *makes at least* $\lfloor \frac{k}{2} \rfloor \frac{m}{24} \log \frac{1}{2\gamma}$ *total queries in expectation.*

We first give the sketch for $k = 2$, and then extend it to $k \geq 2$. For a full proof see Appendix B. **Set-up.** The construction is adapted from Proposition 2 of Thiessen and Gärtner [2021]. Let $e_1, \ldots, e_m$ be the canonical basis of $\mathbb{R}^m$ and let $\ell = \lfloor 1/\sqrt{2\gamma\sqrt{m}} \rfloor$; note that $\gamma \leq \frac{m^{-3/2}}{16}$ and $m \geq 2$ ensure $\ell \geq 4$. Let $p = m - 1$, and for each $i \in [p]$ and $j \in [\ell]$ define $x_i^j = e_i + j \cdot e_m$. Finally, let $X = \{x_i^j : i \in [p], j \in [\ell]\}$ and define the concept class $\mathcal{H} = \left\{ \bigcup_{i \in [p]} \{x_i^1, \ldots, x_i^{\ell_i}\} : (\ell_1, \ldots, \ell_p) \in [\ell]^p \right\}$. Let $\mathcal{C} = (C_1, C_2)$ be any partition of $X$ such that $C_1 \in \mathcal{H}$. One can easily verify that $\mathcal{C}$ has strong convex hull margin $\frac{1}{2\ell^2\sqrt{m}} \geq \gamma$. See Figure 2 for reference.

**Query bound.** Let $V_0 = \{(C_1, C_2) : C_1 \in \mathcal{H}\}$. This is the initial version space. We let the target concept $\mathcal{C} = (C_1, C_2)$ be drawn uniformly at random from $V_0$. Note that for $k = 2$, any lower bound on the number of SEED queries alone, also holds for any combination of SEED and LABEL queries, as LABEL$(x)$ can be simulated by SEED$(x, 1)$. Thus, without loss of generality, we can assume that the algorithm is only using SEED queries. For all $t = 0, 1, \ldots,$ we denote by $V_t$ the version space after the first $t$ SEED queries made by the algorithm. Now fix any $t \geq 1$ and let SEED$(U, y)$ be the $t$-th such query. Without loss of generality we assume $y = 1$; a symmetric argument applies to $y = 2$. If $U \cap C_1$ contains a point $x$ whose label can be inferred from the first $t - 1$ queries, then we return

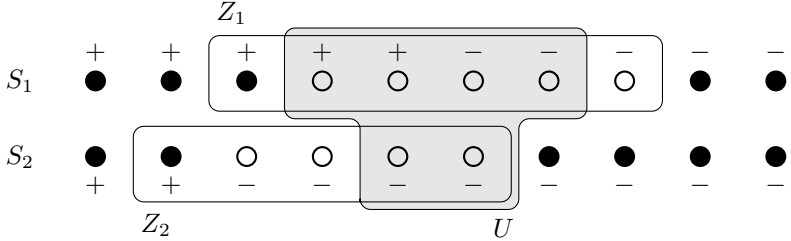

Figure 2: $X$ for $p = 2$ and $\ell = 10$. Filled points represent the agreement region. The maximum point of $S_1 \cap C_1$ (resp. $S_2 \cap C_1$) can be any point in $Z_1$ (resp. $Z_2$). $U$ is a possible query.

$x$. Therefore we can continue under the assumption that $U$ does not contain any such point (doing otherwise cannot reduce the probability that the algorithm learns nothing). The oracle answers so to maximize $\frac{|V_t|}{|V_{t-1}|}$, as described below.

For each $i \in [p]$ let $S_i = \{x_i^j : j \in [\ell]\}$. We consider $S_i$ as sorted by the index $j$. Let $Z_i$ be the subset of $S_i$ in the disagreement region of $V_{t-1}$ together with the point in $S_i$ preceding this region; observe that this point always exists, as $x_i^1 \in C_1$ is in the agreement region. Note that $Z_i$ is necessarily an interval of $S_i$. We let $U_i = Z_i \cap U$ for each $i \in [p]$ and $P(U) = \{i \in [p] : U_i \neq \emptyset\}$. For every $i \in P(U)$, we let $\alpha_i$ be the fraction of points of $Z_i$ that precede the first point in $U_i$. Let $x_i^* = \arg\max\{j : x_i^j \in S_i \cap C_1\}$. Observe that $|V_{t-1}| = \prod_{i \in [p]} |Z_i|$. Indeed, $x_i^*$ is uniformly distributed over $Z_i$; either $x_i^*$ is a point in the disagreement region of $S_i$, or the disagreement region of $S_i$ is fully contained in $C_2$ and $x_i^*$ is the point preceding the disagreement region of $S_i$.

Now we show that $\mathbb{E}[|V_{t-1}|/|V_t|] \leq m$. Let $\mathcal{E}$ be the event that $\text{SEED}(U, 1) = \text{NIL}$. Write:

$$\mathbb{E}\left[\frac{|V_{t-1}|}{|V_t|}\right] = \Pr(\mathcal{E})\,\mathbb{E}\left[\frac{|V_{t-1}|}{|V_t|}\,\bigg|\,\mathcal{E}\right] + \Pr(\overline{\mathcal{E}})\,\mathbb{E}\left[\frac{|V_{t-1}|}{|V_t|}\,\bigg|\,\overline{\mathcal{E}}\right] \qquad (2)$$

We bound each one of the two terms in the right-hand side.

For the first term, note that $\mathcal{E}$ holds if and only if $U_i \cap C_1 = \emptyset$ for all $i \in P(U)$. Since $x_i^*$ is uniformly distributed over $Z_i$, for all $i \in P(U)$ we have $\Pr(C_1 \cap U_i = \emptyset) = \alpha_i$, and since the distributions of those points are independent, then $\Pr(\mathcal{E}) = \prod_{i \in P(U)} \alpha_i$. If $\Pr(\mathcal{E}) > 0$ and $\mathcal{E}$ holds, then $x_i^*$ is uniformly distributed over the first $\alpha_i |Z_i|$ points of $Z_i$, as the rest of $Z_i$ belongs to $C_2$. This holds independently for all $i$, thus:

$$|V_t| = \left(\prod_{i \in P(U)} \alpha_i |Z_i|\right)\left(\prod_{i \in [p] \setminus P(U)} |Z_i|\right) = \left(\prod_{i \in P(U)} \alpha_i\right)\left(\prod_{i \in [p]} |Z_i|\right) = |V_{t-1}| \prod_{i \in P(U)} \alpha_i \quad (3)$$

It follows that $\Pr(\mathcal{E})\mathbb{E}\left[\frac{|V_{t-1}|}{|V_t|}\,\big|\,\mathcal{E}\right] \leq 1$.

Let us turn to the second term. If $\mathcal{E}$ does not hold, then $\text{SEED}(U, 1)$ returns the smallest point $x \in U_i$ for any $i \in P(U)$ such that $C_1 \cap U_i \neq \emptyset$ (note that necessarily $x \in C_1$). For any fixed $i \in P(U)$, the probability of returning the smallest point of $U_i$ is bounded by $\Pr(C_1 \cap U_i \neq \emptyset)$, which is $1 - \alpha_i$; and if this is the case, then we have $|V_t| = (1 - \alpha_i)|V_{t-1}|$. Thus:

$$\Pr(\overline{\mathcal{E}})\mathbb{E}\left[\frac{|V_{t-1}|}{|V_t|}\,\bigg|\,\overline{\mathcal{E}}\right] \leq \Pr(\overline{\mathcal{E}}) \max_{i \in P(U)} (1 - \alpha_i)\frac{1}{(1 - \alpha_i)} = \Pr(\overline{\mathcal{E}}) \leq 1 \qquad (4)$$

So the two terms of (2) are both bounded by 1; we conclude that $\mathbb{E}\left[\frac{|V_{t-1}|}{|V_t|}\right] \leq 2$.

Next, fix any $\bar{t} \geq 1$ and let $\log = \log_2$. By the concavity of $\log$ and by Jensen's inequality:

$$\mathbb{E}\left[\log \frac{|V_0|}{|V_{\bar{t}}|}\right] = \mathbb{E}\left[\sum_{t=1}^{\bar{t}} \log \frac{|V_{t-1}|}{|V_t|}\right] = \sum_{t=1}^{\bar{t}} \mathbb{E}\left[\log \frac{|V_{t-1}|}{|V_t|}\right] \leq \sum_{t=1}^{\bar{t}} \log \mathbb{E}\left[\frac{|V_{t-1}|}{|V_t|}\right] \qquad (5)$$

Since $\mathbb{E}\left[\frac{|V_{t-1}|}{|V_t|}\right] \leq 2$, the right-hand side is at most $\bar{t}$. Now, since $|V_0| = \ell^p = \ell^{m-1}$, by Markov's inequality, and since $(m-1)\log \ell - \log 2 \geq \frac{(m-1)\log \ell}{2} \geq \frac{m \log \ell}{4}$:

$$\Pr(|V_{\bar{t}}| \leq 2) = \Pr\left(\log \frac{|V_0|}{|V_{\bar{t}}|} \geq (m-1)\log \ell - \log 2\right) \leq \frac{4\,\mathbb{E}\left[\log \frac{|V_0|}{|V_{\bar{t}}|}\right]}{m \log \ell} \leq \frac{4\bar{t}}{m \log \ell} \quad (6)$$

Now let $T$ be the random variable counting the number of queries spent by the algorithm, and let $V_T$ be the version space at return time. Since $\mathcal{C}$ is uniform over $V_T$ and $\mathcal{C}$ is returned with probability at least $\frac{1}{2}$, then $\Pr(|V_T| \leq 2) \geq \frac{1}{2}$. By (6) and linearity of expectation,

$$\frac{1}{2} \leq \Pr(|V_T| \leq 2) \leq \sum_{\bar{t} \geq 0} \Pr(T = \bar{t}) \cdot \frac{4\bar{t}}{m \log \ell} = \mathbb{E}[T]\frac{4}{m \log \ell} \quad (7)$$

Therefore $\mathbb{E}[T] \geq \frac{m \log \ell}{4}$. Now, since $\ell \geq 4$ then $\ell \geq \frac{4}{5\sqrt{2\gamma\sqrt{m}}}$, which since $m \leq (16\gamma)^{-2/3}$ yields, after calculations, $\ell \geq \sqrt[3]{1/\gamma} \cdot \frac{4^{4/3}}{5\sqrt{2}} > 0.89 \sqrt[3]{1/\gamma}$. This shows that $E[T] > \frac{m}{24}\log\frac{1}{2\gamma}$, concluding the proof for $k = 2$.

**Extension to k $\geq$ 2.** For each $s \in \lfloor \frac{k}{2} \rfloor$ and each pair of classes $C_{2s-1}, C_{2s}$, use the construction above shifted along the $m$-th dimension by $(s-1)\ell$. One can easily verify that learning $\mathcal{C}$ is as hard as learning $\lfloor \frac{k}{2} \rfloor$ independent binary classifiers, for each of which the bound above holds.

## 5 Conclusions and Future Work

We have shown that, with a careful combination of LABEL and SEED queries, one can overcome the limitations of each query alone and get the "best of both worlds": an algorithm that achieves exponential savings and, simultaneously, has running time polynomial in the dimension of the space. Our work leaves open a few problems. The first problem is to understand the tradeoff between the two query types: how many LABEL does one need if one is allowed only $Q$ SEED? The second problem is whether, for the one-sided case, one can achieve a query rate that is independent of the distortion $\kappa_d$, as we did for the multiclass case. The third problem is whether one can improve the dependence of our bounds on the number $k$ of classes, ideally bounding it by a polynomial.

## Acknowledgments and Disclosure of Funding

The authors gratefully acknowledge partial support by the Google Focused Award "Algorithms and Learning for AI" (ALL4AI). Nicolò Cesa-Bianchi is also supported by the MIUR PRIN grant Algorithms, Games, and Digital Markets (ALGADIMAR) and by the EU Horizon 2020 ICT-48 research and innovation action under grant agreement 951847, project ELISE (European Learning and Intelligent Systems Excellence).

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
