# Active Learning of Classifiers with Label and Seed Queries (Supplementary Material)

**Marco Bressan**
Dept. of CS, Univ. of Milan, Italy
marco.bressan@unimi.it

**Nicolò Cesa-Bianchi**
DSRC & Dept. of CS, Univ. of Milan, Italy
nicolo.cesa-bianchi@unimi.it

**Silvio Lattanzi**
Google
silviol@google.com

**Andrea Paudice**
Dept. of CS, Univ. of Milan, Italy &
Istituto Italiano di Tecnologia, Italy
andrea.paudice@unimi.it

**Maximilian Thiessen**
Research Unit ML, TU Wien, Austria
maximilian.thiessen@tuwien.ac.at

## A    Supplementary material for Section 3

### A.1    Claim 1

**Claim 1.** *Let $K \subset \mathbb{R}^m$ be a convex body, let $E \supseteq K$ be any enclosing ellipsoid, and let $\mu_E$ be the centroid of $E$. Let $f(x) = Ax + \mu$ be an affine transformation with $\|A\|_2 \leq \lambda$ and $\mu \in K$. Then for any $x \in K$ we have $f(x) \in \sigma(E, \mu_E, \lambda + 1)$.*

*Proof.* Without loss of generality, we can assume $K$ to be full rank. We can also assume $E$ to be the $\ell_2$ unit ball; otherwise, just apply an appropriate affine transformation at the beginning of the proof, and its inverse at the end. Under these assumptions, for all $x \in K$ we have $\|x\|_2 \leq 1$, and since $\|\mu\|_2 \leq 1$ as well, we obtain:

$$\|f(x)\|_2^2 = \|Ax\|_2^2 + \|\mu\|_2^2 + 2\langle Ax, \mu \rangle \leq \lambda^2 + 1 + 2\lambda = (\lambda + 1)^2 \tag{8}$$

which implies $f(x) \in (\lambda + 1)E$.                  $\square$

### A.2    Proof of Lemma 8

First, we prove that $E_i \leq m^2(m+1)\operatorname{conv}(C_i)$ for all $i \in [k]$. This is trivial if $E_i = \emptyset$, so assume $E_i \neq \emptyset$ and let $\ell_i \geq 1$ be the value of $h_i$ at return time. For every $h = 1, \ldots, \ell_i$ let $E_i^h = \operatorname{MVE}(S_i^h)$ and let $\mu_i^h$ be the center of $E_i^h$. If $\mu_i$ is the center of $E_i$ then by John's theorem $\sigma\left(E_i, \mu_i, \frac{1}{m}\right) \subseteq \operatorname{conv}(X_i)$, and since $X_i \subset \bigcup_{h=1}^{\ell_i} E_i^h$, then $\operatorname{conv}(X_i) \subseteq \operatorname{conv}\left(\bigcup_{h=1}^{\ell_i} E_i^h\right)$. Moreover $E_i^h \subseteq \sigma\left(\operatorname{conv}(S_i^h), \mu_i^h, m\right)$ for all $h \in [\ell_i]$, which yields:

$$\sigma\left(E_i, \mu_i, \frac{1}{m}\right) \subseteq \operatorname{conv}\bigcup_{h=1}^{\ell_i} \sigma\left(\operatorname{conv}(S_i^h), \mu_i^h, m\right) \tag{9}$$

Thus we need only to show that the right-hand side is in $\sigma(\operatorname{conv}(C_i), \mu, m(m+1))$ for some $\mu \in \mathbb{R}$.

Let $S_i = \cup_{h=1}^{\ell_i} S_i^h$, let $E = \mathrm{MVE}(S_i)$, and let $\mu$ be the center of $E$. (Note that in general $E \neq E_i$). For every $h \in [\ell_i]$, by applying Claim 1 from Appendix A to $f(x) = \sigma(x, \mu_i^h, m)$ and by John's theorem:

$$\sigma\big(\mathrm{conv}(S_i^h), \mu_i^h, m\big) \subseteq \sigma(E, \mu, m+1) \subseteq \sigma(\mathrm{conv}(S_i), \mu, m(m+1)) \tag{10}$$

By taking the union over all $h \in [\ell_i]$, and since $\mathrm{conv}(S_i) \subseteq \mathrm{conv}(C_i)$, we obtain:

$$\bigcup_{h=1}^{\ell_i} \sigma\big(\mathrm{conv}(S_i^h), \mu_i^h, m\big) \subseteq \sigma(\mathrm{conv}(C_i), \mu, m(m+1)) \tag{11}$$

As the right-hand side is a convex set, (11) still holds if the left-hand side is replaced by its own convex hull; but that convex hull is the right-hand side of (9), which proves the sought claim.

We conclude the proof. For the correctness, since $E_i \leq m^2(m+1)\,\mathrm{conv}(C_i)$, and since the updates at lines 1 and 1 guarantee that $(X_i)_{i \in [k]}$ is a partition of $X$, then $((X_i, E_i))_{i \in [k]}$ is an $m^2(m+1)$-rounding of $X$. For the running time, the **for** loops perform $k \leq n$ iterations, and the **while** loop performs at most $n$ iterations as each iteration strictly decreases the size of $X$. The running time of any iteration is dominated by the computation of $\mathrm{MVE}(S_i)$ or $\mathrm{MVE}(X_i)$, which takes time $\mathrm{poly}(n + m)$, see above. Hence $\mathrm{Round}(X, k)$ runs in time $\mathrm{poly}(n + m)$. For the query bounds, the **while** loop makes $\mathcal{O}(m^2 k)$ LABEL queries per iteration. By standard generalization bounds, since the VC dimension of ellipsoids in $\mathbb{R}^m$ is $\mathcal{O}(m^2)$, $E_i^h$ contains at least half of $X \cap C_i$ with probability at least $\frac{1}{2}$, and thus the expected number of rounds before $X$ becomes empty is in $\mathcal{O}(k \lg n)$, see Bressan et al. [2021a]. We conclude that $\mathrm{Round}(X, k)$ uses $\mathcal{O}(m^2 k^2 \lg n)$ LABEL queries in expectation.

### A.3 Pseudocode of CPLearn **and full proof of Theorem 10**

We present CPLearn and prove Theorem 10. The pseudocode of CPLearn is given in Algorithm 4 below; to keep that pseudocode readable we have omitted some details, discussing them in the proof (e.g., the choice of some parameters). For the sake of the proof we suppose $h^{-1}(*) = \emptyset$. It is immediate to verify that the proof holds when $h^{-1}(*) \neq \emptyset$, too, since SEED never returns points in $h^{-1}(*)$ and thus CPLearn behaves identically on $X$ and on $X \setminus h^{-1}(*)$.

CPLearn starts by issuing $\mathrm{SEED}(X, +1)$ and $\mathrm{SEED}(X, -1)$ at lines 4–4, and if either one returns NIL then we immediately return $(\emptyset, X)$ or $(X, \emptyset)$ accordingly, which is clearly correct. Therefore we can continue assuming none of the two queries returned NIL.

**Reduction to the homogeneous case via lifting.** CPLearn works on a *lifted* version of the problem where the target separator is homogeneous. For any $z \in \mathbb{R}^m$ and any $c \in \mathbb{R}$ let $(z, c) \in \mathbb{R}^{m+1}$ be the vector obtained by extending $z$ with a coordinate equal to $c$. For each $x \in X$ let $x' = (x, R)$, and let $X' = \{x' : x \in X\}$ as in lines 4–4. Extend $h$ to $X'$ in the natural way by defining $h(x') = h(x)$ for any $x' \in X'$. We claim that $\{x' \in X' : h(x') = +1\}$ and $\{x' \in X' : h(x') = -1\}$ are separated in $\mathbb{R}^{m+1}$ by a homogeneous hyperplane with margin $\frac{r}{2}$. To see this, let $u \in S^{m-1}$ and $b \in \mathbb{R}$ such that $h(x) \cdot (\langle x, u \rangle + b) \geq r$ for all $x \in X$ with $h(x) \neq *$; such $u$ and $b$ exist by the assumptions of the theorem, and note that $b \leq R$. Now let $v = (u, b/R)$ and let $u' = \frac{v}{\|v\|_2}$; note that $\|v\|_2 \leq \|u\|_2 + \frac{b}{R} \leq 2$. Then, for every $x' \in X'$:

$$\langle x', u' \rangle = \frac{\langle x', v \rangle}{\|v\|_2} = \frac{\langle x, u \rangle + R \cdot b/R}{\|v\|_2} = \frac{\langle x, u \rangle + b}{\|v\|_2} \tag{12}$$

which implies:

$$h(x') \cdot \langle x', u' \rangle = \frac{h(x) \cdot (\langle x, u \rangle + b)}{\|v\|_2} \geq \frac{r}{\|v\|_2} \geq \frac{r}{2} \tag{13}$$

Thus the problem of learning $h$ reduces to learning the lifted version of $h$ over $X'$, which is realized by a homogeneous separator with margin $\frac{r}{2}$. The rest of the proof shows that CPLearn from line 4 onward solves this lifted problem under the bounds of Theorem 10.

**Overview.** At a high level, CPLearn is a cutting-plane algorithm—see, e.g., Mitchell [2003]. Starting with $V_0$ being the $(m+1)$-dimensional unit ball $B(0, 1)$, CPLearn computes a sequence of version

---

**Algorithm 4:** CPLearn$(X)$

---

**if** SEED$(X, +1) = $ NIL **then return** $(\emptyset, X)$
**if** SEED$(X, -1) = $ NIL **then return** $(X, \emptyset)$
$R \leftarrow \max_{x \in X} \|x\|_2$
$X' \leftarrow \{(x, R) : x \in X\}$
$i \leftarrow 0, V_0 \leftarrow B(0, 1)$ in $\mathbb{R}^{m+1}$
**for** $i \leftarrow 0, \ldots, n$ **do**
> draw $N = \Theta(m^6 n^{2a})$ points $z_1, \ldots, z_N$ independently $\frac{1}{2m^3}$-uniformly at random from $V_i$
> $\hat{\mu}_i \leftarrow \frac{1}{N} \sum_{j=1}^{N} z_j$
> $X'_i \leftarrow \{x' \in X' : \langle \hat{\mu}_i, x' \rangle \geq 0\}$
> $X_i \leftarrow$ projection of $X'_i$ on $\mathbb{R}^m$
> **if** SEED$(X_i, -1) = $ NIL *and* SEED$(X \setminus X_i, +1) = $ NIL **then**
>> **return** $(X_i, X \setminus X_i)$
>
> **else**
>> exclude all points returned by the queries from future queries
>> let $u_i$ be any point returned by either query
>> **if** $i = 0$ **then**
>>> $u_i^* \leftarrow u_i$
>>
>> **else**
>>> $u_i^* \leftarrow u_i - z_0 \cdot \frac{\langle u_i, \hat{\mu}_i \rangle}{\langle z_0, \hat{\mu}_i \rangle}$ where $z_0 = h(u_0) \cdot u_0$
>>
>> $V_{i+1} \leftarrow V_i \cap \{x' \in \mathbb{R}^{m+1} : h(u_i) \cdot \langle u_i^*, x' \rangle \geq 0\}$
>
> draw points independently near-uniformly at random from $V_i$ until $N = \mathrm{poly}(m + n)$ of them, $z_1, \ldots, z_N$, fall in $V_{i+1}$
> use the covariance matrix of $\{z_1, \ldots, z_N\} \cap V_{i+1}$ to compute a coordinate system under which $V_{i+1}$ is $t$-rounded

---

spaces $V_1, V_2, \ldots$ by setting $V_{i+1} = V_i \cap Z_i^*$, where $Z_i^*$ is some halfspace determined through SEED queries, as follows. For every $i \geq 0$ let $\mu_i$ be the center of mass of $V_i$, and consider the halfspace:

$$H_i = \{x' \in \mathbb{R}^{m+1} : \langle \mu_i, x' \rangle \geq 0\} \tag{14}$$

Now let $X'_i = X' \cap H_i$ and execute SEED$(X'_i, -1)$ and SEED$(X' \setminus X'_i, +1)$. If both return NIL then clearly $(X_i, X \setminus X_i)$, where $X_i$ is the projection of $X'_i$ on $\mathbb{R}^m$, is the partition of $X$ induced by $h$. If instead either query returns a point $u_i$, then consider the halfspace:

$$Z_i = \{x' \in \mathbb{R}^{m+1} : h(u_i) \cdot \langle u_i, x' \rangle \geq 0\} \tag{15}$$

Finally, let $V_{i+1} = V_i \cap Z_i$ and repeat. By standard arguments, $\mathrm{vol}(V_{i+1}) \leq (1 - 1/e) \mathrm{vol}(V_i)$ but $V_{i+1}$ contains a ball of radius $\Omega(r/R)$, and the process terminates within $\mathcal{O}(m \log \frac{R}{r})$ iterations, see for instance [Gilad-Bachrach et al., 2004, Theorem 2].

There are two main obstacles in implementing this process. The first obstacle is computing $\mu_i$, which is hard in general [Rademacher, 2007]. Fortunately, we can efficiently compute a point $\hat{\mu}_i$ that with good probability yields the same guarantees as $\mu_i$, by sampling from a near-uniform distribution over $V_i$ via the hit-and-run random walk technique of Lovász and Vempala [2006]. The second obstacle is that, in order for hit-and-run to be efficient, we must have a system of coordinates under which $V_i$ is well-rounded, i.e., not "too thin" along any direction. Unfortunately, letting $V_{i+1} = V_i \cap Z_i$ may make $V_{i+1}$ extremely thin, as we have no control over $Z_i$ (it depends on the SEED answers). Therefore, CPLearn carefully rotates $Z_i$ into a new halfspace $Z_i^*$ such that $V_{i+1} = V_i \cap Z_i^*$ contains $V_i \cap Z_i$, and that $\mathrm{vol}(V_i \cap Z_i^*)$ is not much smaller than $\mathrm{vol}(V_i)$. This allows CPLearn to sample efficiently from $V_{i+1}$; using those samples it then computes a coordinate system under which $V_{i+1}$ is again well-rounded.

**A complete proof.** We say a convex body $K \subset \mathbb{R}^{m+1}$ is $t$-rounded if $B(0, t) \subseteq K \subseteq B(0, 1)$. For every $u \in \mathbb{R}^{m+1}$ let $h_u = \{x \in X : \langle u, x' \rangle \geq 0\}$. Fix $t \in \Omega(1/m)$ and $c > 0$ sufficiently small, and fix $a > 0$ arbitrarily large. We show an implementation of CPLearn that satisfies the following invariants:

1. $V_i$ contains all vectors $u \in \mathbb{R}^{m+1}$ such that $h_u = h$
2. $\mathrm{vol}(V_{i+1}) \le (1 - c)\,\mathrm{vol}(V_i)$
3. $V_i$ is $t$-rounded under the coordinate system currently held by CPLearn

We prove that the first invariant holds deterministically for all $i \ge 0$, and that with probability at least $1 - n^{1-a}$ the other ones hold for all $i \ge 0$. Together with the argument from Gilad-Bachrach et al. [2004] recalled above, the first two invariants imply that CPLearn returns a separator of $X$ w.r.t. $h$ in $\mathcal{O}(m \log \frac{R}{r})$ iterations (and thus SEED queries). The third invariant ensures that CPLearn can sample enough points from the version space $V_i$ in time $\mathrm{poly}(n + m)$, which in turn ensures the overall running time is in $\mathrm{poly}(n + m)$, where the degree depends on $a$.

Let us first discuss how at lines 4 and 4 one can sample from $V_i$ and $V_{i+1}$ in time $\mathrm{poly}(n + m)$ per sample, assuming both $V_i$ and $V_{i+1}$ are $t$-rounded in the coordinate system held by CPLearn. Let $K$ be a $t$-rounded convex body in $\mathbb{R}^{m+1}$. For any given $\epsilon > 0$, the hit-and-run algorithm of Lovász and Vempala [2006] returns a point $\epsilon$-uniformly at random from $K$ after $\mathcal{O}(m^3 t^2 \ln t/\epsilon)$ steps; see Corollary 1.2 of Lovász and Vempala [2006]. Moreover, every step of that algorithm can be implemented in time polynomial in the representation of $K$, see for instance Bressan et al. [2021a]. By letting $K = V_i$, and noting that the representation of $V_i$ has size $\mathcal{O}(m + n)$ as $i \le n$ and every constraining halfspace can be encoded in $\mathcal{O}(m)$ bits, we can sample a point $\epsilon$-uniformly in time $\mathrm{poly}(n, m, \ln t/\epsilon)$ per sample; the same holds for $V_{i+1}$. Since we set $t = \Omega(1/m)$ and $\epsilon = \Omega(1/\mathrm{poly}(n + m))$, we conclude that lines 4 and 4 take $\mathrm{poly}(n + m)$ time per sample.

Let us now turn to the invariants. Consider first the case $i = 0$. The first and third invariant hold trivially, while the second one holds for any $c \le 1/2$ since $V_1$ is the intersection of $V_0 = B(0, 1)$ and a homogeneous halfspace. Let then $i \ge 1$ and suppose all invariants hold for $i - 1$. We prove that they hold for $i + 1$ as well.

Let $\eta = 1/2m^2$, let $\epsilon = \frac{\eta}{m}$, and $p = n^{-a}/2$. Then, line 4 draws $N = \Theta(m^2/\eta^2 p^2)$ independent $\epsilon$-uniform random points $z_1, \dots, z_N$ from $V_i$, and line 4 sets $\hat{\mu}_i$ as their average. As shown in Bressan et al. [2021a], this implies $\Pr(d(\hat{\mu}_i, \mu_i) \le \eta \phi(V_i)) \ge 1 - p$, where $\phi(V_i)$ is the Euclidean diameter of $V_i$. As $V_i$ is $t$-rounded, $\phi(V_i) \le 2$, hence $\Pr(d(\hat{\mu}_i, \mu_i) \le 1/m^2) \ge 1 - n^{-a}/2$. Now suppose indeed $d(\hat{\mu}_i, \mu_i) \le 1/m^2$. It is not hard to see that any halfspace $Z$ containing $\hat{\mu}_i$ satisfies $\mathrm{vol}(Z \cap V_i) \ge \frac{1}{e}(1 - \frac{1}{m})^{m+1}\,\mathrm{vol}(V_i) = \Omega(\mathrm{vol}(V_i))$; that is, $\hat{\mu}_1$ has Tukey depth at least $c$ (see the second invariant).

Next, consider the set $X_i'$ computed at line 4, and observe that $X_i' = X \cap H_i$, where:

$$H_i = \{x' \in \mathbb{R}^{m+1} : \langle \hat{\mu}_i, x' \rangle \ge 0\} \tag{16}$$

Clearly, if the two queries at line 4 return NIL, then CPLearn returns the correct partition of $X$. Otherwise consider the point $u_i$ returned by either query, see line 4, and let $Z_i$ as in (15). By standard arguments $\hat{\mu}_i \in Z_i$, and therefore $\mathrm{vol}(V_i \cap Z_i) \le (1 - c)\,\mathrm{vol}(V_i)$ as said above. Moreover, again by standard arguments, $V_i \cap Z_i$ contains all vectors $u \in \mathbb{R}^{m+1}$ such that $h_u = h$.

Now let us turn to CPLearn. Since $i \ge 1$, CPLearn at line 4 defines:

$$u_i^* = u_i - z_0 \cdot \frac{\langle u_i, \hat{\mu}_i \rangle}{\langle z_0, \hat{\mu}_i \rangle} \tag{17}$$

Before continuing, we check that $u_i^*$ is well-defined, i.e., that $\langle z_0, \hat{\mu}_i \rangle > 0$. Indeed, $\hat{\mu}_i$ lies in the interior of $V_i$ since it has positive Tukey depth (see above), and since by construction $V_i \subseteq Z_0$ for all $i \ge 1$, then $\hat{\mu}_i$ lies in the interior of $Z_0$ too. Moreover $z_0$ lies in the interior of $Z_0$, too, being the normal vector of $Z_0$. Hence $\langle z_0, \hat{\mu}_i \rangle > 0$, as claimed. Note also that, for every $x \in \mathbb{R}^{m+1}$, the definition of $u_i^*$ and the linearity of the inner product yield:

$$\langle u_i^*, x \rangle = \langle u_i, x \rangle - \langle z_0, x \rangle \cdot \frac{\langle u_i, \hat{\mu}_i \rangle}{\langle z_0, \hat{\mu}_i \rangle} \tag{18}$$

Now, CPLearn at line 4 sets $V_{i+1} = V_i \cap Z_i^*$, where:

$$Z_i^* = \{x \in \mathbb{R}^{m+1} : h(u_i) \cdot \langle u_i^*, x \rangle \ge 0\} \tag{19}$$

We are now ready to prove the three invariants above.

*The first invariant.* We claim that $V_i \cap Z_i \subseteq V_i \cap Z_i^*$. In fact, we claim $Z_0 \cap Z_i \subseteq Z_0 \cap Z_i^*$; this implies $V_i \cap Z_i \subseteq V_i \cap Z_i^*$, since by construction $V_i \subseteq Z_0$ as $i \ge 1$. In turn, since $V_i \cap Z_i$ contains

all vectors $u \in \mathbb{R}^{m+1}$ such that $h_u = h$, see above, this implies that $V_{i+1}$ contains all those vectors as well, proving the first invariant. Let $x \in Z_0 \cap Z_i$. Then:

$$h(u_i) \cdot \langle u_i^*, x \rangle = h(u_i) \cdot \langle u_i, x \rangle - h(u_i) \cdot \langle z_0, x \rangle \cdot \frac{\langle u_i, \hat{\mu}_i \rangle}{\langle z_0, \hat{\mu}_i \rangle} \tag{20}$$

Let us examine the terms of (20). First, $h(u_i) \cdot \langle u_i, x \rangle \geq 0$ since $x \in Z_i$. Second, $\langle z_0, x \rangle \geq 0$ since $x \in Z_0$. Third, $\langle z_0, \hat{\mu}_i \rangle > 0$ as noted above. Thus the term $-h(u_i) \cdot \langle z_0, x \rangle \cdot \frac{\langle u_i, \hat{\mu}_i \rangle}{\langle z_0, \hat{\mu}_i \rangle}$ has the same sign as $-h(u_i) \cdot \langle u_i, \hat{\mu}_i \rangle$. However, by definition $u_i$ is a counterexample to the labeling given by $H_i$, which means $h(u_i) \cdot \langle u_i, \hat{\mu}_i \rangle < 0$. Therefore $h(u_i) \cdot \langle u_i^*, x \rangle \geq 0$, which implies $x \in Z_i^*$ as desired.

*The second invariant.* We claim that $\hat{\mu}_i \in Z_i^*$. To this end just substitute $x = \hat{\mu}_i$ in (18) to see that $\langle u_i^*, \hat{\mu}_i \rangle = 0$. since $\mu_i$ has Tukey depth $c > 0$ w.r.t. $V_i$, we deduce that $\text{vol}(V_{i+1}) = \text{vol}(V_i \cap Z_i^*) \leq (1 - c)\,\text{vol}(V_i)$. This proves the second invariant.

*The third invariant.* First of all, we claim that $\text{vol}(V_{i+1}) = \text{vol}(V_i \cap Z_i^*) \geq c\,\text{vol}(V_i)$. To this end just observe that $\hat{\mu}_i$ is on the boundary of $\mathbb{R}^{m+1} \setminus Z_i^*$, too. Consider then line 4 of CPLearn: if the samples are independent $\epsilon$-uniform over $V_i$, then every sample drawn ends in $V_{i+1}$ independently with probability at least $c - \epsilon$. Hence, as long as $\epsilon < c/2$, a sample of $\Theta(N)$ such points from $V_i$ will contain a subsample of $N$ points $z_1, \ldots, z_N$ in $V_{i+1}$ with probability $1 - e^{-\Theta(N)}$. Moreover, those $N$ samples will be $\frac{\epsilon}{c}$-uniform in $V_{i+1}$. Therefore line 4 takes time $\text{poly}(n + m)$ with probability $1 - e^{-\text{poly}(n+m)}$. For $N$ large enough, the inverse of the covariance matrix of $z_1, \ldots, z_N$ CPLearn yields a coordinate system under which $V_{i+1}$ is $t$-rounded with probability at least $1 - n^{-a}/2$, see for instance Vempala [2010]. This proves the third invariant.

**Wrap-up.** Note that CPLearn makes at most $n$ iterations, as every iteration either returns (if the SEED queries return NIL) or decreases the number of points of $X$ for which the label is not known (see line 4). Hence, with probability at least $1 - n^{1-a}$, all the invariants above hold for all $i = 0, \ldots, n-1$. The query bounds and the running time bounds follow as explained above.

### A.4 One-sided margin

We sketch the proof of Theorem 3. Let $d$ be a metric over $\mathbb{R}^m$ induced by some norm $\| \cdot \|_d$. We say $C \subseteq X$ has one-sided strong convex hull margin $\gamma$ with respect to $d$ if $d(\text{conv}(X \setminus C), \text{conv}(C)) \geq \gamma \phi_d(C)$.

The idea behind Theorem 3 is to compute a Euclidean *one-sided $\alpha$-rounding* of $X$ w.r.t. $h$, that is, a set $\widehat{X} \subseteq X$ such that $C \subseteq \widehat{X}$ and $\widehat{X} \leq \alpha \text{conv}(C)$, where $C = h^{-1}(+1)$. We will compute $\widehat{X}$ for $\alpha = \text{poly}\left(\frac{\kappa_d}{\gamma}\right)$, and then use the cutting-planes algorithm of Section 3.2. As the margin is invariant under scaling, assume without loss of generality $\inf_{u \in S^{m-1}} \|u\|_d = 1$ and $\sup_{v \in S^{m-1}} \|v\|_d = \kappa_d$. Let $x = \text{SEED}(X, +1)$. If $x = \text{NIL}$ then clearly $h = -1$. Otherwise we run $\text{BallSearch}(X, x)$, listed below. BallSearch sorts $X$ by distance from $x$, and then uses LABEL queries to perform a binary search and find a pair of points $x_\text{lo} \in C$ and $x_\text{hi} \in X \setminus C$ adjacent in the ordering. (This works even if the order is not monotone w.r.t. the labels). At this point BallSearch guesses a value $t$ for $\frac{\gamma}{\kappa_d}$, starting with $t = 1$. Given $t$, with a SEED query BallSearch checks if there are points of $C$ among the points at distance between $d_\text{euc}(x, x_\text{hi})$ and $\frac{1}{t}d_\text{euc}(x, x_\text{hi})$ from $x_\text{hi}$. If not, then it lets $\widehat{X} = X \cap B(x, d_\text{euc}(x, x_\text{lo}))$, else it lets $\widehat{X} = X \cap B(x, \frac{1}{t}d_\text{euc}(x, x_\text{hi}))$. Finally, it checks whether $C \subseteq \widehat{X}$; if yes then it returns $\widehat{X}$, else it halves $t$ and repeat. One can show that this procedure stops with $t \geq \frac{\gamma}{2\kappa_d}$, yielding a $\widehat{X}$ such that $\phi(\widehat{X}) = \mathcal{O}(\phi(C)/t)$ and that $C$ and $\widehat{X} \setminus C$ are linearly separated with margin $\Omega\left(t\frac{\gamma}{\kappa_d}\phi(\widehat{X})\right)$. Setting $R = \phi(\widehat{X})$ and $r = d_\text{euc}(C, \widehat{X} \setminus C)$, we conclude that $\frac{R}{r} = \text{poly}\left(\frac{\kappa_d}{\gamma}\right)$. At this point by Theorem 10 we can compute $C$ by running $\text{CPLearn}(\widehat{X})$, which takes time $\text{poly}(n + m)$ and uses $\mathcal{O}\left(m \log \frac{\kappa_d}{\gamma}\right)$ SEED queries in expectation.

**A remark on Theorem 3.** Given two pseudometrics $d$ and $q$ induced by seminorms $\| \cdot \|_d$ and $\| \cdot \|_q$, let $\kappa_d(q) = \sup_{u \in S_q^{m-1}} \|u\|_d / \inf_{v \in S_q^{m-1}} \|v\|_d$. If one can compute $\| \cdot \|_q$ efficiently, then Theorem 3 holds with $\kappa_d(q)$ in place of $\kappa_d$. In fact, Theorem 3 is just the special case where $q = d_\text{euc}$. Therefore one can restate Theorem 3 so that $d$ is an arbitrary pseudometric (thus including the case $\kappa_d = \infty$), provided one has access to an approximation $q$ of $d$ with finite distortion.

**Algorithm 5:** BallSearch$(X, x_1)$

---

let $x_1, \ldots, x_n$ be the points of $X$ in order of Euclidean distance from $x_1$ (break ties arbitrarily)
**if** LABEL$(x_n) = +1$ **then return** $X$
lo $\leftarrow 1$, hi $\leftarrow n$
**while** hi $-$ lo $\geq 2$ **do**
$\quad$ $i \leftarrow \lceil \frac{\text{hi} + \text{lo}}{2} \rceil$
$\quad$ **if** LABEL$(x_i) = 1$ **then** lo $\leftarrow i$ **else** hi $\leftarrow i$
$t \leftarrow 1$, $r \leftarrow d_{\text{euc}}(x_1, x_{\text{lo}})$, $R \leftarrow d_{\text{euc}}(x_1, x_{\text{hi}})$
**repeat**
$\quad$ $U_i \leftarrow \{x \in X : R \leq d_{\text{euc}}(x, x_1) \leq \frac{1}{t}R\}$
$\quad$ **if** SEED$(U_i, +1) = $ NIL **then** $\widehat{X} \leftarrow X \cap B(x_1, r)$ **else** $\widehat{X} \leftarrow X \cap B\left(x_1, \frac{1}{t}R\right)$
$\quad$ $t \leftarrow t/2$
**until** SEED$(X \setminus \widehat{X}, +1) = $ NIL
**return** $\widehat{X}$;

---

# B  Supplementary material for Section 4

## B.1  Full proof of Theorem 4

**Construction.** We first discuss the case $k = 2$. Let $e_1, \ldots, e_m$ be the canonical basis of $\mathbb{R}^m$. To ease the notation define $p = m - 1$; the input set will span a $p$-dimensional subspace. Define:

$$\ell = \left\lfloor \frac{1}{\sqrt{2\gamma\sqrt{m}}} \right\rfloor \tag{21}$$

Since $\gamma \leq \frac{m^{-3/2}}{16}$ and $m \geq 2$,

$$\ell \geq \frac{1}{\sqrt{2\frac{m^{-3/2}}{16}\sqrt{m}}} = \sqrt{8m} \geq 4 \tag{22}$$

For each $i \in [p]$ and $j \in [\ell]$, let $x_i^j = e_i + j \cdot e_m$. Finally, let $X = \{x_i^j : i \in [p], j \in [\ell]\}$. Define the concept class:

$$\mathcal{H} = \left\{ \bigcup_{i \in [p]} \{x_i^1, \ldots, x_i^{\ell_i}\} \; : \; (\ell_1, \ldots, \ell_p) \in [\ell]^p \right\} \tag{23}$$

Let $\mathcal{C} = \{C_1, C_2\}$ be any partition of $X$ with $C_1 \in \mathcal{H}$ and $C_2 = X \setminus C_1$. First, we observe that $C_1$ and $C_2$ are separated by a hyperplane. Let $(\ell_1, \ldots, \ell_p)$ be the vector defining $C_1$. Then we let:

$$u = (-\ell_1, \ldots, -\ell_p, 1) \tag{24}$$

Then for any $x_i^j \in X$,

$$\langle u, x_i^j \rangle = -\ell_i + j \tag{25}$$

which is bounded from above by zero if and only if $j \leq \ell_i$, that is, if and only if $x_i^j \in C_1$. Hence $C_1$ and $C_2$ admit a linear separator. Next we prove that, under the Euclidean distance, $C_1$ and $C_2$ have strong convex hull margin $\gamma$. Using the vector $u$ defined above, since every $x_i^j \in C_2$ has $j \geq \ell_i + 1$, then $\langle u, x_i^j \rangle \geq 1$. This implies:

$$d(\text{conv}(C_1), \text{conv}(C_2)) \geq \frac{1}{\|u\|_2} \geq \frac{1}{\sqrt{p\ell^2 + 1}} \geq \frac{1}{\ell\sqrt{m}} \tag{26}$$

The diameter of $C_1$ is at most that of $X$, which equals $d(x_1^1, x_2^\ell) \leq \ell - 1 + \sqrt{2} \leq 2\ell$. Together with (26) and the fact that $\ell \leq \frac{1}{\sqrt{2\gamma\sqrt{m}}}$, this provides:

$$d(\text{conv}(C_1), \text{conv}(C_2)) \geq \frac{1}{2\ell^2\sqrt{m}} \phi_d(C_1) \geq \frac{2\gamma\sqrt{m}}{2\sqrt{m}} \phi_d(C_1) = \gamma\,\phi_d(C_1) \tag{27}$$

The same holds for $C_2$. Hence $\mathcal{C}$ has strong convex hull margin $\gamma$.

**Query bound.** Let $V_0 = \{(C_1, C_2) : C_1 \in \mathcal{H}\}$. This is the initial version space. We let the target concept $\mathcal{C} = (C_1, C_2)$ be drawn uniformly at random from $V_0$. For all $t = 0, 1, \ldots$, we denote by $V_t$ be the version space after the first $t$ SEED queries made by the algorithm. Now fix any $t \geq 1$ and let SEED$(U, y)$ be the $t$-th such query. Without loss of generality we assume $y = 1$; a symmetric argument applies to $y = 2$. If $U \cap C_1$ contains a point $x$ in the agreement region of $V_{t-1}$, i.e., whose label can be inferred from past queries, then we return $x$. Therefore we can continue under the assumption that $U$ does not contain any such point (doing otherwise cannot reduce the probability that the algorithm learns nothing). The oracle answers so to maximize $\frac{|V_t|}{|V_{t-1}|}$, as described below.

For each $i \in [p]$ let $S_i = \{x_i^j : j \in [\ell]\}$. We consider $S_i$ as a sequence of points sorted by the index $j$. Let $Z_i$ be the subset of $S_i$ in the disagreement region of $V_{t-1}$ together with the point in $S_i$ preceding this region; observe that this point always exists, as $x_i^1 \in C_1$ is in the agreement region. Note that $Z_i$ is necessarily an interval of $S_i$. We let $U_i = Z_i \cap U$ for each $i \in [p]$ and $P(U) = \{i \in [p] : U_i \neq \emptyset\}$. For every $i \in P(U)$, we let $\alpha_i$ be the fraction of points of $Z_i$ that precede the first point in $U_i$. Let $x_i^* = \arg\max\{j : x_i^j \in S_i \cap C_1\}$. Observe that $|V_{t-1}| = \prod_{i \in [p]} |Z_i|$, as $x_i^*$ can be every point of $Z_i$. Indeed, $x_i^*$ is uniformly distributed over $Z_i$; either $x_i^*$ is a point in the disagreement region of $S_i$, or the disagreement region of $S_i$ is fully contained in $C_2$ and $x_i^*$ is the point preceding the disagreement region of $S_i$.

Now we show that $\mathbb{E}[|V_{t-1}|/|V_t|] \leq p + 1$. Let $\mathcal{E}$ be the event that SEED$(U, 1) = $ NIL. Write:

$$\mathbb{E}\left[\frac{|V_{t-1}|}{|V_t|}\right] = \Pr(\mathcal{E}) \, \mathbb{E}\left[\frac{|V_{t-1}|}{|V_t|} \,\Big|\, \mathcal{E}\right] + \Pr(\overline{\mathcal{E}}) \, \mathbb{E}\left[\frac{|V_{t-1}|}{|V_t|} \,\Big|\, \overline{\mathcal{E}}\right] \tag{28}$$

We bound the two terms of (28) starting with the first one. Note that $\mathcal{E}$ holds if and only if $U_i \cap C_1 = \emptyset$ for all $i \in P(U)$. Since $x_i^*$ is uniformly distributed over $Z_i$, for all $i \in P(U)$ we have:

$$\Pr(C_1 \cap U_i = \emptyset) = \alpha_i \tag{29}$$

And since the distributions of those points are independent:

$$\Pr(\mathcal{E}) = \prod_{i \in P(U)} \Pr(C_1 \cap U_i = \emptyset) = \prod_{i \in P(U)} \alpha_i \tag{30}$$

If $\Pr(\mathcal{E}) > 0$ and $\mathcal{E}$ holds, then $x_i^*$ is uniformly distributed over the first $\alpha_i |Z_i|$ points of $Z_i$, as the rest of $Z_i$ belongs to $C_2$. This holds independently for all $i$, thus:

$$|V_t| = \left(\prod_{i \in P(U)} \alpha_i |Z_i|\right)\left(\prod_{i \in [p] \setminus P(U)} |Z_i|\right) = \left(\prod_{i \in P(U)} \alpha_i\right)\left(\prod_{i \in [p]} |Z_i|\right) = |V_{t-1}| \prod_{i \in P(U)} \alpha_i \tag{31}$$

It follows that $\Pr(\mathcal{E}) \mathbb{E}\left[\frac{|V_{t-1}|}{|V_t|} \,\Big|\, \mathcal{E}\right] \leq 1$.

Let us now bound the second term of (28). If $\mathcal{E}$ does not hold, then SEED$(U, 1)$ returns the smallest point $x \in U_i$ for any $i \in P(U)$ such that $C_1 \cap U_i \neq \emptyset$ (note that necessarily $x \in C_1$). For any fixed $i \in P(U)$, the probability of returning the smallest point of $U_i$ is bounded by $\Pr(C_1 \cap U_i \neq \emptyset)$, which is $1 - \alpha_i$; and if this is the case, then we have $|V_t| = (1 - \alpha_i)|V_{t-1}|$. Thus:

$$\Pr(\overline{\mathcal{E}}) \mathbb{E}\left[\frac{|V_{t-1}|}{|V_t|} \,\Big|\, \overline{\mathcal{E}}\right] \leq \Pr(\overline{\mathcal{E}}) \max_{i \in P(U)} (1 - \alpha_i)\frac{1}{(1 - \alpha_i)} = \Pr(\overline{\mathcal{E}}) \leq 1 \tag{32}$$

So the two terms of (2) are both bounded by 1; we conclude that $\mathbb{E}\left[\frac{|V_{t-1}|}{|V_t|}\right] \leq 2$.

We can conclude the query bound. For any $\bar{t} \geq 1$,

$$\mathbb{E}\left[\log \frac{|V_0|}{|V_{\bar{t}}|}\right] = \mathbb{E}\left[\sum_{t=1}^{\bar{t}} \log \frac{|V_{t-1}|}{|V_t|}\right] \tag{33}$$

$$= \sum_{t=1}^{\bar{t}} \mathbb{E}\left[\log \frac{|V_{t-1}|}{|V_t|}\right] \tag{34}$$

$$\leq \sum_{t=1}^{\bar{t}} \log \mathbb{E}\left[\frac{|V_{t-1}|}{|V_t|}\right] \qquad \text{Jensen's inequality} \tag{35}$$

$$\leq \sum_{t=1}^{\bar{t}} \log 2 \qquad \text{see above} \tag{36}$$

$$= \bar{t} \tag{37}$$

Since $|V_0| = \ell^{m-1}$, by Markov's inequality, and since $(m-1)\log \ell - \log 2 \geq \frac{(m-1)\log \ell}{2} \geq \frac{m \log \ell}{4}$:

$$\Pr(|V_{\bar{t}}| \leq 2) = \Pr\left(\log \frac{|V_0|}{|V_{\bar{t}}|} \geq (m-1)\log \ell - \log 2\right) \leq \frac{4\,\mathbb{E}\left[\log \frac{|V_0|}{|V_{\bar{t}}|}\right]}{m \log \ell} \leq \frac{4\,\bar{t}}{m \log \ell} \tag{38}$$

Now let $T$ be the random variable counting the number of queries spent by the algorithm, and let $V_T$ be the version space at return time. Since $\mathcal{C}$ is uniform over $V_T$ and $\mathcal{C}$ is returned with probability at least $\frac{1}{2}$, then $\Pr(|V_T| \leq 2) \geq \frac{1}{2}$. By (38) and linearity of expectation,

$$\frac{1}{2} \leq \Pr(|V_T| \leq 2) = \sum_{\bar{t} \geq 0} \Pr(T = \bar{t})\Pr(|V_{\bar{t}}| \leq 2) \leq \sum_{\bar{t} \geq 0} \Pr(T = \bar{t}) \cdot \frac{4\bar{t}}{m \log \ell} = \mathbb{E}[T]\frac{4}{m \log \ell} \tag{39}$$

Therefore $\mathbb{E}[T] \geq \frac{m \log \ell}{8}$. Now, since $\ell \geq 4$ then $\ell \geq \frac{4}{5\sqrt{2\gamma\sqrt{m}}}$, which since $m \leq (16\gamma)^{-2/3}$ yields

$$\ell \geq \frac{4}{5\sqrt{2\gamma(16\gamma)^{-1/3}}} = \sqrt[3]{\frac{1}{\gamma}}\frac{4}{5\sqrt{2(16)^{-1/3}}} = \sqrt[3]{\frac{1}{\gamma}}\frac{4 \cdot 4^{1/3}}{5\sqrt{2}} \tag{40}$$

Since $\frac{4^{4/3}}{5\sqrt{2}} > 0.89$, we conclude that:

$$\mathbb{E}[T] > \frac{m \log \frac{0.89}{\sqrt[3]{\gamma}}}{8 \log m} > \frac{m\frac{1}{3}\log \frac{1}{2\gamma}}{8 \log m} = \frac{m \log \frac{1}{2\gamma}}{24 \log m} \tag{41}$$

which concludes the proof for $k = 2$.

**Multiclass.** For any $k \geq 2$ let $k' = \left\lfloor \frac{k}{2} \right\rfloor$. For each $s \in [k']$ consider the construction for the case $k = 2$ shifted along the $m$-th dimension by $(s-1)\ell \cdot e_m$:

$$X_s = \left\{x_i^j + (s-1)\ell \cdot e_m : i \in [p], j \in [\ell]\right\} \tag{42}$$

We let $X^* = \bigcup_{s \in [k']} X_s$, and we define the possible subsets of $X^*$ corresponding to class $C_{2s-1}$ as:

$$\mathcal{H}_s = \left\{\bigcup_{i \in [p]} \left\{x_i^1 + (s-1)\ell \cdot e_m,\ \ldots,\ x_i^{\ell_i} + (s-1)\ell \cdot e_m\right\}\ :\ (\ell_1, \ldots, \ell_p) \in [\ell]^p\right\} \tag{43}$$

Finally, let $\mathcal{H}$ be the set of all partitions $\mathcal{C} = (C_1, \ldots, C_k)$ of $X^*$ such that $C_{2s-1} \in \mathcal{H}_s$ and $C_{2s} = X_s \setminus C_{2s-1}$ for all $s \in [k']$, and let $C_k = \emptyset$ in case $k$ is odd. The same arguments of the case $k = 2$ prove that any such $\mathcal{C}$ has convex hull margin $\gamma$. Indeed, for adjacent classes $C_i, C_{i+1}$ those arguments prove that the strong convex hull margin is at least $\gamma$; for non-adjacent classes, the margin can only be larger. The random target concept $\mathcal{C} = (C_1, \ldots, C_k)$ is obtained by drawing each $C_{2s-1}$ for $s \in [k']$ uniformly at random from $\mathcal{H}_s$, and letting $C_{2s} = X_s \setminus C_{2s-1}$.

We turn to the bound. Consider a generic query $\textsc{seed}(U, i)$ issued by the algorithm. Without loss of generality we can assume $U \subseteq C_{2s-1} \cup C_{2s} = X_s$ where $s = \lfloor \frac{i}{2} \rfloor$; indeed, by construction of $\mathcal{H}$, that query can never return a point in $U \setminus X_s$. This shows that learning $\mathcal{C}$ requires solving the $k'$ independent binary instances $X_s$, returning $\mathcal{C}_s = (C_{2s-1}, C_{2s})$, for $s \in [k']$. As the probability of returning $\mathcal{C}$ is bounded from above by the minimum over $s \in [k]$ of the probability of returning $\mathcal{C}_s$, the algorithm must make at least $\frac{m}{24} \log \frac{1}{2\gamma}$ queries for each $s \in [k']$, concluding the proof.

## C  Supplementary material for Section A.4

**Lemma 13.** *Let $C \subseteq X$ have strong convex hull margin $\gamma \in (0, 1]$ w.r.t. d. For any $x_1 \in C$* $\text{BallSearch}(X, x_1)$ *takes time* $\text{poly}(n + m)$*, uses* $\mathcal{O}(\log n)$ LABEL *queries and* $\mathcal{O}(\log \frac{\kappa_d}{\gamma})$ SEED *queries, and outputs* $\widehat{X} \subseteq X$ *such that*

1. $C \subseteq \widehat{X}$
2. $d_{\text{euc}}(\text{conv}(C), \text{conv}(\widehat{X} \setminus C)) \geq \frac{\gamma^2}{4\kappa_d^2} \phi(\widehat{X})$

*Proof.* To begin, observe that $d_{\text{euc}} \leq d \leq \kappa_d\, d_{\text{euc}}$ implies that the ratio between distances changes by a factor at most $\kappa_d$ between $d_{\text{euc}}$ and $d$. In particular this implies that for any set $\widehat{X} \subseteq X$:

$$\frac{d_{\text{euc}}(\text{conv}(C), \text{conv}(\widehat{X} \setminus C))}{\phi(C)} \geq \frac{d(\text{conv}(C), \text{conv}(\widehat{X} \setminus C))}{\kappa_d\, \phi_d(C)} \tag{44}$$

We will use this inequality below.

Now, suppose line 5 of BallSearch returns, so $\widehat{X} = X$. The running time, the query bounds, and point (1) are straightforward. To prove (2), since $x_1, x_n \in C$ we have:

$$\phi(C) \geq d_{\text{euc}}(x_1, x_n) \geq \frac{1}{2}\phi(X) = \frac{1}{2}\phi(\widehat{X}) \geq \frac{\gamma}{2\kappa_d}\phi(\widehat{X}) \tag{45}$$

where we used $\phi(X) = \max_{a,b \in X} d_{\text{euc}}(a, b) \leq \max_{a,b \in X}(d_{\text{euc}}(a, x_1) + d_{\text{euc}}(x_1, b)) \leq 2d_{\text{euc}}(x_1, x_n)$. Therefore $\phi(\widehat{X}) \leq \frac{2\kappa_d}{\gamma}\phi(C)$, which together with (44) and the margin condition gives:

$$\frac{d(\text{conv}(C), \text{conv}(\widehat{X} \setminus C))}{\phi(\widehat{X})} \geq \frac{d_{\text{euc}}(\text{conv}(C), \text{conv}(\widehat{X} \setminus C))}{\frac{2\kappa_d}{\gamma}\phi(C)} \geq \frac{d(\text{conv}(C), \text{conv}(\widehat{X} \setminus C))}{\frac{2\kappa_d}{\gamma}\kappa_d\, \phi_d(C)} \geq \frac{\gamma^2}{2\kappa_d^2} \tag{46}$$

We turn to the **repeat** loop. Consider a generic iteration just before the update of $t$. We prove:

(a) $d(C, \widehat{X} \setminus C) \geq \min\left(t, \frac{\gamma}{\kappa_d}\right)\frac{\gamma}{2\kappa_d}\phi(\widehat{X})$

(b) if $t \leq \frac{\gamma}{\kappa_d}$ then $C \subseteq \widehat{X}$

First, suppose $\textsc{seed}(U_i, +1) = \textsc{nil}$, in which case $\widehat{X} = X \cap B(x_1, r)$. To prove (a), observe that $x_1, x_{\text{lo}} \in C$ implies:

$$\phi(C) \geq d_{\text{euc}}(x_1, x_{\text{lo}}) = r \geq \frac{1}{2}\phi(\widehat{X}) \geq \min\left(\frac{t}{2}, \frac{\gamma}{2\kappa_d}\right)\phi(\widehat{X}) \tag{47}$$

Now use the argument above, but with $1/\min\left(\frac{t}{2}, \frac{\gamma}{2\kappa_d}\right)$ in place of $\frac{2\kappa_d}{\gamma}$ in (46). To prove (b), note that $x_1 \in C$ and $x_{\text{hi}} \in X \setminus C$ implies $R = d_{\text{euc}}(x_1, x_{\text{hi}}) \geq d_{\text{euc}}(C, X \setminus C)$. Since $d_{\text{euc}} \leq d \leq \kappa_d\, d_{\text{euc}}$, and by the margin assumptions,

$$\frac{R}{\phi(C)} \geq \frac{d_{\text{euc}}(C, X \setminus C)}{\phi(C)} \geq \frac{d(C, X \setminus C)}{\kappa_d\, \phi_d(C)} \geq \frac{\gamma}{\kappa_d} \geq \min\left(t, \frac{\gamma}{\kappa_d}\right) \tag{48}$$

Therefore $\phi(C) \le \max\left(\frac{1}{t}, \frac{\kappa_d}{\gamma}\right)R$, which implies $C \subseteq X \cap B\left(x_1, \max\left(\frac{1}{t}, \frac{\kappa_d}{\gamma}\right)R\right)$. For $t \le \frac{\kappa_d}{\gamma}$ the right-hand side is $X \cap B(x_1, \frac{1}{t}R)$. Note however that $X \cap B(x_1, \frac{1}{t}R) = (X \cap B(x_1, r)) \cup U_i$ since $x_{\mathrm{lo}}, x_{\mathrm{hi}}$ are adjacent in the sorted list. But $\mathrm{SEED}(U_i, +1) = \mathrm{NIL}$, hence $C \subseteq X \cap B(x_1, r) = \widehat{X}$.

Next, suppose $\mathrm{SEED}(U_i, +1) = y \ne \mathrm{NIL}$, in which case $\widehat{X} = X \cap B(x_1, \frac{1}{t}R)$. To prove (a), note that $\phi(C) \ge d(x_1, y) \ge R$, and that $\phi(\widehat{X}) \le 2\frac{1}{t}R$. Hence $\phi(C) \ge \frac{t}{2}\phi(\widehat{X}) \ge \min\left(\frac{t}{2}, \frac{\gamma}{2\kappa_d}\right)\phi(\widehat{X})$. Now use again the argument above, but with $1/\min\left(\frac{t}{2}, \frac{\gamma}{2\kappa_d}\right)$ in place of $\frac{2\kappa_d}{\gamma}$ in (46). To prove (b), the argument for the case above implies $C \subseteq X \cap B\left(x_1, \max\left(\frac{1}{t}, \frac{\kappa_d}{\gamma}\right)R\right)$. If $t \le \frac{\gamma}{\kappa_d}$ then the right-hand side is just $\widehat{X}$.

To conclude the proof, note that by point (b) above the **repeat** loop returns in $\mathcal{O}(\log \frac{\kappa_d}{\gamma})$ iterations. Therefore $\mathrm{BallSearch}(X, x_1)$ uses $\mathcal{O}(\log n)$ LABEL queries and $\mathcal{O}(\log \frac{\kappa_d}{\gamma})$ SEED queries. Finally, note that the running time can be brought to $\mathrm{poly}(n + m)$ by storing the output of all SEED queries, and replacing $U_i$ with $U_i \setminus U_i \cap \hat{C}$ where $\hat{C} \subset C$ is the subset of points of $C$ known so far. In this way, at each **repeat** iteration either $\widehat{X}_i \subseteq C$ or we learn the label of some point of $C$ previously unknown. Therefore **repeat** makes at most $n$ iterations; it is immediate to see that each iteration takes time $\mathrm{poly}(n + m)$ and thus $\mathrm{BallSearch}$ runs in time $\mathrm{poly}(n + m)$ as well. $\qquad\square$

### C.1 Proof of Theorem 3

Let $x = \mathrm{SEED}(X, +1)$. If $x = \mathrm{NIL}$ then stop and return $\emptyset$. Otherwise run $\mathrm{BallSearch}(X, x)$ to obtain $\widehat{X}$. By Lemma 13 this takes $\mathrm{poly}(n + m)$ time, $\mathcal{O}(\log n)$ LABEL queries, and $\mathcal{O}(\log \frac{\kappa_d}{\gamma})$ SEED queries. By Lemma 13 $C \subseteq \widehat{X}$, and $C$ and $\widehat{X} \setminus C$ are linearly separated with margin $\frac{\gamma^2}{4\kappa_d^2}\phi(\widehat{X})$. Thus $\widehat{X}$ satisfies the assumptions of Theorem 10 with $R/r = \frac{4\kappa_d^2}{\gamma^2}$, and by running $\mathrm{CPLearn}(\widehat{X})$ we obtain $C$ in time $\mathrm{poly}(n + m)$ using $\mathcal{O}(m \log \frac{\kappa_d}{\gamma})$ SEED queries in expectation.

## D  Bounds for inputs with bounded bit complexity

We consider the case where $X$ has bounded bit complexity, distinguishing two widely used cases.

### D.1  Rational coordinates

Supose $X \subset \mathbb{Q}^m$ and every $x \in X$ can be encoded in $b(x) \le B$ bits as follows [Korte and Vygen, 2018]. If $x \in \mathbb{Z}$, then $b(x) = 1 + \lceil \log(|x| + 1)\rceil$. If $x = p/q \in \mathbb{Q}$ with $p, q \in \mathbb{Z}$ coprime, then $b(x) = b(p) + b(q)$. If $x \in \mathbb{Q}^m$, then $b(x) = m + \sum_{i \in [m]} b(x_i)$. We show that $B$ gives a lower bound on the margin. The argument is related to Kwek and Pitt [1998].

**Lemma 14.** *Suppose $X \subset \mathbb{Q}^m$ has bit complexity bounded by $B$, and suppose $C \subseteq X$ and $X \setminus C$ are linearly separable. Then $d(\mathrm{conv}(C), \mathrm{conv}(X \setminus C)) \ge 2^{-\mathcal{O}(m^2 B)}$.*

*Proof.* Let $P = \mathrm{conv}(C)$ and let $H$ be a hyperplane containing a face of $P$. By Lemma 4.5 of Korte and Vygen [2018], $H = \{x \in \mathbb{R}^m : \langle w, x\rangle = t\}$ for some $w \in \mathbb{Q}^m$ and $t \in \mathbb{Q}$ such that $b(w) + b(t) \le 75m^2 B$. The distance between $H$ and any $x \in X \setminus C$ is:

$$d(x, H) = \frac{|\langle w, x\rangle - t|}{\|w\|_2} \tag{49}$$

To bound $|\langle w, x\rangle - t|$ suppose $w, x, t$ are encoded by:

$$w_i = \frac{p_w^i}{q_w^i} \quad i \in [m], \qquad x_i = \frac{p_x^i}{q_x^i} \quad i \in [m], \qquad t = \frac{p_t}{q_t} \tag{50}$$

Replacing those quantities in the expression of $|\langle w, x\rangle - t|$, taking the common denominator, observing that the numerator of the resulting expression is an integer, and recalling that $|\langle w, x\rangle - t| > 0$, we deduce:

$$|\langle w, x\rangle - t| \ge \frac{1}{q_t \prod_{i \in [m]} q_w^i q_x^i} \tag{51}$$

However, since $b(x) = \mathcal{O}(\log(1 + |x|))$ for any $x \in \mathbb{Z}$,

$$b\left(q_t \prod_{i \in [m]} q_w^i q_x^i\right) = \mathcal{O}\left(b(q_t) + \sum_{i \in [m]} (b(w_i) + b(x_i))\right) = \mathcal{O}(b(t) + b(w) + b(x)) \qquad (52)$$

which therefore is in $\mathcal{O}(m^2 B)$. Therefore $|\langle w, x \rangle - t| \geq 2^{-\mathcal{O}(m^2 B)}$. To bound $\|w\|_2$ we just note that $\|w\|_2 \leq \|w\|_1 \leq 2^{b(w)} \leq 2^{75m^2 B}$. We conclude that:

$$d(x, H) = \frac{|\langle w, x \rangle - t|}{\|w\|_2} \geq 2^{-\mathcal{O}(m^2 B)} \qquad (53)$$

The proof is complete. □

**Corollary 15.** *Suppose $X \subset \mathbb{N}^m$ has bit complexity bounded by $B \in \mathbb{N}$ in the rational coordinates model, and let $\mathcal{C} = (C_1, \ldots, C_k)$ be a partition of $X$ such that $C_i, C_j$ are linearly separable for every distinct $i, j \in [k]$. Then $\mathcal{C}$ can be learned in time $\mathrm{poly}(n + m)$ using $\mathcal{O}(k^2 m^3 B)$ SEED queries in expectation.*

*Proof.* Any $x \in X$ satisfies $\|x\|_2 \leq \|x\|_1 \leq 2^B$, and by Lemma 14 any two distinct classes $C_i, C_j \in \mathcal{C}$ are linearly separable with margin $r = 2^{-\mathcal{O}(m^2 B)}$. By Theorem 10, CPLearn$(X)$ with SEED restricted to classes $i, j$ returns a separator for $C_i$ and $C_j$ in time $\mathrm{poly}(m + n)$ using $\mathcal{O}(m \log \frac{R}{r}) = \mathcal{O}(m^3 B)$ SEED queries in expectation. By intersecting the separators for all $j \in [k] \setminus i$ we obtain $C_i$. Repeating this process for all $i \in [k]$ yields the claim. □

### D.2 Grid

Let $c > 0$ be such that $1/c$ is an integer and suppose that $X \subseteq Q = \{-1, -1 + c, \ldots, 1 - c, 1\}^m$. We call this the grid model. If $1/c \leq 2^{B/m} - 1$ then we say that the bit complexity of $X$ is bounded by $B$.

**Corollary 16.** *Suppose $X \subset \mathbb{N}^m$ has bit complexity bounded by $B \in \mathbb{N}$ in the grid model, and let $\mathcal{C} = (C_1, \ldots, C_k)$ be a partition of $X$ such that $C_i, C_j$ are linearly separable for every distinct $i, j \in [k]$. Then $\mathcal{C}$ can be learned in time $\mathrm{poly}(n + m)$ using $\mathcal{O}(k^2 m(B + \log m))$ SEED queries in expectation.*

*Proof.* We use the approach of Gonen et al. [2013]. Let $c > 0$ be such that $1/c$ is an integer and suppose that $X \subseteq Q = \{-1, -1 + c, \ldots, 1 - c, 1\}^m$. By Lemma 10 of Gonen et al. [2013], any two sets in $Q$ that are linearly separable are also linearly separable with margin $r = (c/\sqrt{m})^{m+2}$. We can thus apply CPLearn as in the proof of Corollary 15, obtaining for separating every $C_i, C_j$ a running time of $\mathrm{poly}(m + n)$ and an expected query bound of $\mathcal{O}(m \log \frac{R}{r}) = \mathcal{O}(m^2 \log(m/c))$. Since $c \geq 2^{-B/m} - 1$, then the bound becomes $\mathcal{O}(m^2 \log(m 2^{B/m})) = \mathcal{O}(m^2(B/m + \log m)) = \mathcal{O}(m(B + \log m))$. This proves the total expected query bound of $\mathcal{O}(k^2 m(B + \log m))$. □