# OpenReview forum: "Active Learning of Classifiers with Label and Seed Queries"
_NeurIPS.cc/2022/Conference — NeurIPS 2022 Accept_

### Official Review · Reviewer_Z1n9 · 2022-07-11

**Rating:** 5
**Confidence:** 1
**Soundness:** 3 good
**Presentation:** 4 excellent
**Contribution:** 3 good

**Summary:**

This paper studies active learning of classifiers with margin (assumption of strong convex hull margin), by combining label and seed queries. Using seed queries only, the target classifier can be learned in $\mathcal{O}(m\text{log}n)$ -- for a set of n points in $\mathbb{R}^m$ -- but the algorithm has superpolynomial time complexity. The paper shows that using both label and seed queries, a binary classifier can be learned in $\text{poly}(n+m)$ time, using $\mathcal{O}(m \frac{\text{log}m}{\gamma})$ seed queries, where $\gamma$ is the strong convex hull margin and $\mathcal{O}(m^2 \text{log}n)$ label queries. These results are extended for multi-class classifiers with $k! k^2$ multiplicative overhead. The authors also show that the results hold even when only one class has strong convex hull margin, or when input points have bounded bit complexity.

**Questions:**

The results and some algorithms seem novel, however I am not familiar with the area, so I cannot assess the significance fully.

**Limitations:**

The assumptions are stated clearly and there are no negative direct negative societal impacts.

**Strengths And Weaknesses:**

The paper is written and presented well, the assumptions are stated well, and it solves a well motivated issue of efficiently leveraging seed queries, by combining with label queries. It also provides a novel efficient algorithm (CPLearn) for seed which provides the upper bound on seed queries.

---

> ### Author Response · Authors · 2022-08-02
> **Reply by the authors**
>
> We thank the reviewer for appreciating our contribution even though it does not fall into their area of expertise.

---

### Official Review · Reviewer_TPhg · 2022-07-12

**Rating:** 7
**Confidence:** 3
**Soundness:** 4 excellent
**Presentation:** 4 excellent
**Contribution:** 3 good

**Summary:**

This paper tackles the problem of active learning on classifiers with strong convex hull margin classes. In this framework that generalizes SVM margins, it is known that the query complexity of learning with only label queries is exponential in the dimension and that learning with seed queries is computationally inefficient. Is designed in this paper an approach that combines label and seed queries to learn such classifiers with a low query complexity in polynomial time. In addition to the upper bound on the query complexity of the provided algorithm, is established here a worst case lower bound on the query complexity of learning a k-class classifier with strong convex hull margin.

**Questions:**

For the query complexities, it seems like the lower bound is linear in the dimension while the upper bound for the label query is quadratic.
- Which of the two could be improved?
- Are there interesting cases where the upper and lower bounds match?
- Is there a problem dependent quantity that characterizes the problem?

Are the algorithms BinLearn and KClassLearn implementable in practice?


**Limitations:**

The assumptions made are adequately stated. It would be nice to more clearly address the limitations of the work: without empirical support the algorithms provided in this work seem more like proofs of concept and it feels like there are still some open problems on the query complexity upper bounds. The authors state that the potential negative societal impacts of their work is N/A due to its theoretical nature. It might still be valuable to mention what could go wrong if the suggested algorithms were actually deployed.

**Strengths And Weaknesses:**

Strengths:
- Presentation: the problem is well introduced and the main results are clearly presented
- Impact: some of the results established seem to be of general interest (in particular CPLearn’s guarantees) in addition to solving the problem of learning through label and seed queries with a computationally efficient algorithm whose total query complexity is polynomial in the dimension of the space.
- The paper is technically sound.

Weaknesses:
- No experimental results limit the impact of the work.
- It would be nice to see some mentions of future work and remaining open problems

---

> ### Author Response · Authors · 2022-08-02
> **Reply by the authors**
>
> R: “No experimental results limit the impact of the work.”
>
> A: Our work has a theoretical goal: understanding whether, by carefully combining two types of query, one can bypass the well-known computational and information-theoretic limitations of each of them. We believe we fulfill that goal, and we do not see the lack of experiments as a weakness. This is in line with related works, see e.g. (Kane et al., 2017) or (Hopkins et al., 2021). Moreover, we believe that implementing our algorithm leaves so much room for engineering and tuning (recall that it uses cutting planes, hit-and-run, isotropic position estimation, ...) that any reasonable experimental evaluation would be a work on its own.
>
> R: “It would be nice to see some mentions of future work and remaining open problems.”
>
> A: Thank you for the suggestion. The main open problems are (1) closing the gap between upper and lower bounds, and (2) providing a tradeoff between label and seed complexity (that is: if we are allowed at most S seed queries, how many label queries do we need?). We will add them to the final version.
>
> R: “For the query complexities, it seems like the lower bound is linear in the dimension while the upper bound for the label query is quadratic. Which of the two could be improved?”
>
> A: Actually, our bounds are somewhat tight: if $\gamma = O(n^{-m})$, then the lower bound is in $\Omega(m^2 \log n)$ and matches the label part of the upper bound; and if $\gamma = O(m^{-a})$ for some constant $a > 0$, then the lower bound is in $\Omega(m \log (m/\gamma))$ and matches the seed part of the upper bound. One could explicit this by writing the lower bounds as a $\min(\ldots)$, but we preferred to keep it light. We will however add a remark.
>
> R: “Are there interesting cases where the upper and lower bounds match?”
>
> A: One such case is the class of instances given by the lower bound (Theorem 4); we do not know if that can be deemed interesting though. We have not found other particularly interesting cases yet.
>
> R: “Is there a problem dependent quantity that characterizes the problem?”
>
> A: Our bounds are parameterized by $n, m, k, \gamma$, so they are already instance-dependent in some sense. But we agree that developing an even more refined (and possibly tight) instance-dependent parameterization would be interesting.
>
> R: “Are the algorithms BinLearn and KClassLearn implementable in practice?”
>
> A: Definitely yes, if one can implement label and seed queries (see our answer to Reviewer 4ffc). The only tricky part in our opinion is the near-uniform sampler for convex bodies used by CPLearn. In our proof we use hit-and-run because it has crisp theoretical guarantees, but in practice one can use any efficient sampler of choice; and since such samplers are used by standard LP solvers, our algorithm is implementable as much as those LP solvers are.
>
> R: “It might still be valuable to mention what could go wrong if the suggested algorithms were actually deployed.”
>
> A: We believe there are no specific risks involved in deploying our algorithms. The potential risks are comparable to those of any other general-purpose active learning technology.

---

### Official Review · Reviewer_4ffc · 2022-07-12

**Rating:** 7
**Confidence:** 3
**Soundness:** 3 good
**Presentation:** 3 good
**Contribution:** 3 good

**Summary:**

This paper considers learning of k-class classifiers using label and seed queries, where seed queries are special ones that ask for some labeled examples in the specified region. The paper proposes an algorithm which learns the concepts in polynomial time, where the complexity involves convex-hull margin, an extension of margin in linear classes. The paper also shows a lower bound of necessary queries which shows the tightness of the analysis.


**Questions:**

Are there any practical scenarios where seed queries are realizable?

**Strengths And Weaknesses:**

Strength:
- strong theory paper with a lower bound



Weakness:
- no experiments is done

---

> ### Author Response · Authors · 2022-08-02
> **Reply by the authors**
>
> R: “no experiments is done”
>
> A: Our work has a theoretical goal: understanding whether, by carefully combining two types of query, one can bypass the well-known computational and information-theoretic limitations of each of them. We believe we fulfill that goal, and we do not see the lack of experiments as a weakness. This is in line with related works, see e.g. (Kane et al., 2017) or (Hopkins et al., 2021). Moreover, we believe that implementing our algorithm leaves so much room for engineering and tuning (recall that it uses cutting planes, hit-and-run, isotropic position estimation, ...) that any reasonable experimental evaluation would be a work on its own.
>
> R: “Are there any practical scenarios where seed queries are realizable?”
>
> A: Yes. For instance, seed queries can be realized efficiently when outlier detection is possible. This is the case for vision and cybersecurity, where accurate out-of-distribution tests are available.

---

### Author Response · Authors · 2022-08-02
**Rebuttal**

We thank all reviewers for their constructive comments. We would like to point out that we have improved the lower bound of Theorem 4 by removing the $\log m$ factor at the denominator. This makes our bounds tight for some range of the parameters, as we explain in the replies.

---

### Meta-Review · Area_Chair_YUu9 · 2022-08-30

**Recommendation:** Accept
**Confidence:** Certain

**Metareview:**

The problem is well introduced and the main results are clearly presented
While there are no experimental results, the theoretical contribution seems strong.
Please add the main open problems to the final version.



**Award:**

No

---

### Decision · Program_Chairs · 2022-09-14

Accept